# Astrocyte plasticity in mice ensures continued endfoot coverage of cerebral blood vessels following injury and declines with age

William A. Mills III[1,2,3,4], AnnaLin M. Woo[1,2,9], Shan Jiang [5,6,9], Joelle Martin[4], Dayana Surendran [1,2], Matthew Bergstresser[7], Ian F. Kimbrough [1,2], Ukpong B. Eyo [1,2,3], Michael V. Sofroniew[8] & Harald Sontheimer[1,2 ✉]

Astrocytes extend endfeet that enwrap the vasculature, and disruptions to this association which may occur in disease coincide with breaches in blood-brain barrier (BBB) integrity. Here we investigate if focal ablation of astrocytes is sufficient to disrupt the BBB in mice. Targeted two-photon chemical apoptotic ablation of astrocytes induced a plasticity response whereby surrounding astrocytes extended processes to cover vascular vacancies. In young animals, replacement processes occur in advance of endfoot retraction, but this is delayed in aged animals. Stimulation of replacement astrocytes results in constriction of pre-capillary arterioles, suggesting that replacement astrocytes are functional. Pharmacological inhibition of pSTAT3, as well as astrocyte specific deletion of pSTAT3, reduces astrocyte replacement post-ablation, without perturbations to BBB integrity. Similar endfoot replacement occurs following astrocyte cell death due to reperfusion in a stroke model. Together, these studies uncover the ability of astrocytes to maintain cerebrovascular coverage via substitution from nearby cells.

[1] Brain, Immunology, and Glia Center, University of Virginia School of Medicine, Charlottesville, VA, USA. [2] Department of Neuroscience, University of Virginia School of Medicine, Charlottesville, VA, USA. [3] Robert M. Berne Cardiovascular Research Center, University of Virginia School of Medicine, Charlottesville, VA, USA. [4] Graduate Program in Translational Biology, Medicine, & Health, Virginia Polytechnic Institute and State University, Blacksburg, VA, USA. [5] Department of Material Science and Engineering, Stanford University, Stanford, CA, USA. [6] Wu Tsai Neurosciences Institute, Stanford University, Stanford, CA, USA. [7] School of Neuroscience, Virginia Polytechnic Institute and State University, Blacksburg, VA, USA. [8] Department of Neurobiology, University of California, Los Angeles, CA, USA. [9] These authors contributed equally: AnnaLin M. Woo, Shan Jiang. ✉email: nkn7mv@virginia.edu

Astrocytes serve essential roles in supporting normal brain physiology[1]. This is made possible, in part, by the extension of large, flattened processes, called endfeet, that wrap around blood vessels. Thought to cover up to ~99% of the cerebrovascular surface[2], astrocytic endfeet, in conjunction with pericytes[3], helps to maintain expression of molecules that form the blood-brain barrier (BBB)—including endothelial tight junction, enzymatic, and transporter proteins[4–6]. Astrocytic endfeet also mediates neurovascular coupling, also known as functional hyperemia, whereby local blood flow adjusts to local energy demand. Astrocytes sense changes in neuronal activity via glutamatergic[7] and purinergic receptors that cause increases in $[Ca^{2+}]$. This leads to the release of vasoactive molecules onto pericytes at capillaries[8,9] and arterioles[10–13], leading to changes in vessel diameter.

Interestingly, a number of CNS diseases are marked by retraction or separation of astrocytic endfeet from blood vessels—a phenotype often simultaneously presenting with vascular deficits such as altered BBB permeability or elevated CSF-to-serum albumin ratio, which is indicative of BBB breakdown. Examples include multiple sclerosis[14], major depressive disorder[15–17], ischemia[18–20], and even normal biological aging[21–23]. We previously demonstrated separation of endfeet from the vasculature due to invading glioma cells[24] as well as due to amyloid accumulation on vessels[25]. Both conditions resulted in disruption to neurovascular coupling and, in the case of glioma, BBB breakdown. This raises the question of whether astrocyte endfeet are required to maintain an intact BBB, or whether lost endfeet can be replaced by other astrocytes as has been shown for pericytes[26]. Moreover, since changes in astrocyte morphology and function are known to occur with physiological changes of the organism—i.e., parturition, lactation, chronic dehydration, starvation, voluntary exercise or sleep deprivation[27–30]—it is possible that astrocyte association with blood vessels is equally dependent on the physiological context.

Given the multitude of conditions marked by regions of abnormal vasculature lacking endfoot coverage, we were interested in determining whether replacement endfeet have functional relevance in maintaining BBB integrity and astrocyte-vascular coupling. Using multiphoton imaging through a cranial window, we were able to induce single-cell apoptosis using the targeted two-photon chemical apoptotic ablation (2Phatal) method[31] to question whether loss of endfeet on blood vessels would be compensated for by neighboring cell(s). We find remarkable plasticity, discovering that the ablation of single astrocytes reliably causes innervation by neighboring cells. In young animals, this happens in advance of the ablated cell completely retrieving its process; yet in 12-month-old animals, replacement occurs with a significant 1–2 h delay after the ablated cell has vacated the vessel. Endfoot replacement engages the EGFR/STAT3 signaling pathway, as pharmacological inhibition via AG490 injection was found to impair replacement. Once in place, the replacement endfeet has the ability to vasoconstrict precapillary arterioles like normal astrocytes. Despite recent evidence that global astrocyte loss results in impairments of the BBB[32], we did not find this to be the case even when replacement endfoot coverage was impaired by inhibition of STAT3 phosphorylation. Finally, we demonstrate using focal photothrombosis that astrocyte apoptosis following reperfusion triggers a focal gliovascular plasticity response wherein astrocyte-vascular coverage is maintained. Together, these results reveal a novel process in which astrocytes cover for neighboring cells to maintain vascular coverage in a pSTAT3-dependent manner.

## Results

**Focal ablation of single astrocytes does not breach the BBB but induces an astrocyte endfoot replacement response.** Given our previous finding of compromised BBB integrity in regions of focal endfoot separation due to invading glioma cells, we questioned if the focal ablation of astrocytes is sufficient to induce breaches in BBB integrity. To do so, we implanted cranial windows in mice and adopted the targeted two-photon chemical apoptotic ablation (2Phatal) method developed by Hill et al.[31]. This technique employs the focal illumination properties of a femtosecond-pulsed laser to activate the nucleic-acid binding Hoechst dye (Fig. 1a), triggering apoptosis. Given the ability to induce single-cell apoptosis, we were able to image astrocytes in Aldh1l1-eGFP mice up to the removal of their corpse, which included the retraction of their endfeet (Fig. 1b). We also employed Alexa Fluor 633 hydrazide (Supplementary Fig. 1a, b) to selectively label arterioles[33]. Though we observed no reduction in NG2-dsRed pericyte cell volume or changes in capillary diameter upon single astrocyte ablation in Aldh1l1-eGFP x NG2-dsRed mice (Supplementary Fig. 2a–j), we wanted to avoid any transcriptional perturbations to pericyte physiology, which are known to play a role in BBB integrity and vasodilation of capillaries[34,35]. Comparing the extravasation of retro-orbitally injected 3 kDa TRITC at the time of endfoot retraction to baseline revealed no apparent difference in BBB permeability (Fig. 1c–f). 3 kDa TRITC was chosen due to initial attempts to use the ~1 kDa Cadaverine revealing that at baseline, this dye extravasates and is taken up by astrocytes (Supplementary Fig. 3a). Positive control experiments utilizing direct laser irradiation of the vasculature demonstrated that we were able to detect leakage of 3 kDa TRITC from the vasculature (Supplementary Fig. 3c–g).

At the initial stages of astrocyte endfoot retraction, we observed nearby neighboring astrocytes extend processes to the soon-to-be vacancy left by the ablated astrocyte (Fig. 1e). Given that astrocyte endfeet have been reported to cover up to 99% of the entire cerebrovascular surface[36], we then asked if this process occurred at all levels of the vascular tree. To answer this, Alexa Fluor 633 hydrazide was again employed to specifically label arterioles, and Alexa 633 negative vessels larger than 10 μm were identified as venules. All vessels smaller than 10 μm were identified as capillaries (Supplementary Fig. 1). 2Phatal ablation of astrocytes revealed that the replacement of endfeet also occurred at capillaries (Fig. 1g–i) and venules (Fig. 1k–n), and we henceforth refer to this process of astrocytes extending processes to innervate (cover) vacant vascular regions as "endfoot plasticity".

Because we ablated in the range of two to five astrocytes in the prior experiment—we next sought to determine if ablating one astrocyte was sufficient to induce a similar plasticity response from nearby surrounding astrocytes. We performed this only at arterioles and stratified our analysis into two groups based upon the morphological relationship of the original astrocyte to the blood vessel. It is known that astrocyte somas occupy vascular locations on arterioles and venules, whereas contact from astrocytes at capillaries typically occurs solely by endfeet[37]. We refer to those astrocytes whose somas occupy vascular territory as vessel-associated astrocytes (VAA) (Supplementary Fig. 4A), and those whose somas lie in parenchymal space as parenchymal-associated astrocytes (PAA) (Supplementary Fig. 4B, C). Ablating only one VAA or one PAA successfully elicited a plasticity response from surrounding astrocytes (Supplementary Fig. 4d–g). Given that astrocytes do not move in response to injury[38], it can be assumed that replacement endfoot vascular coverage would be less for ablated VAAs than those astrocytes providing vascular coverage with their endfeet exclusively (PAAs). Quantifying the area of eGFP+ endfoot lining at arterioles pre- versus post-ablation revealed that this is indeed the case (Supplementary Fig. 4h–j). Interestingly, post-ablation area values of eGFP+ endfoot vessel lining for both VAA and PAA were significantly

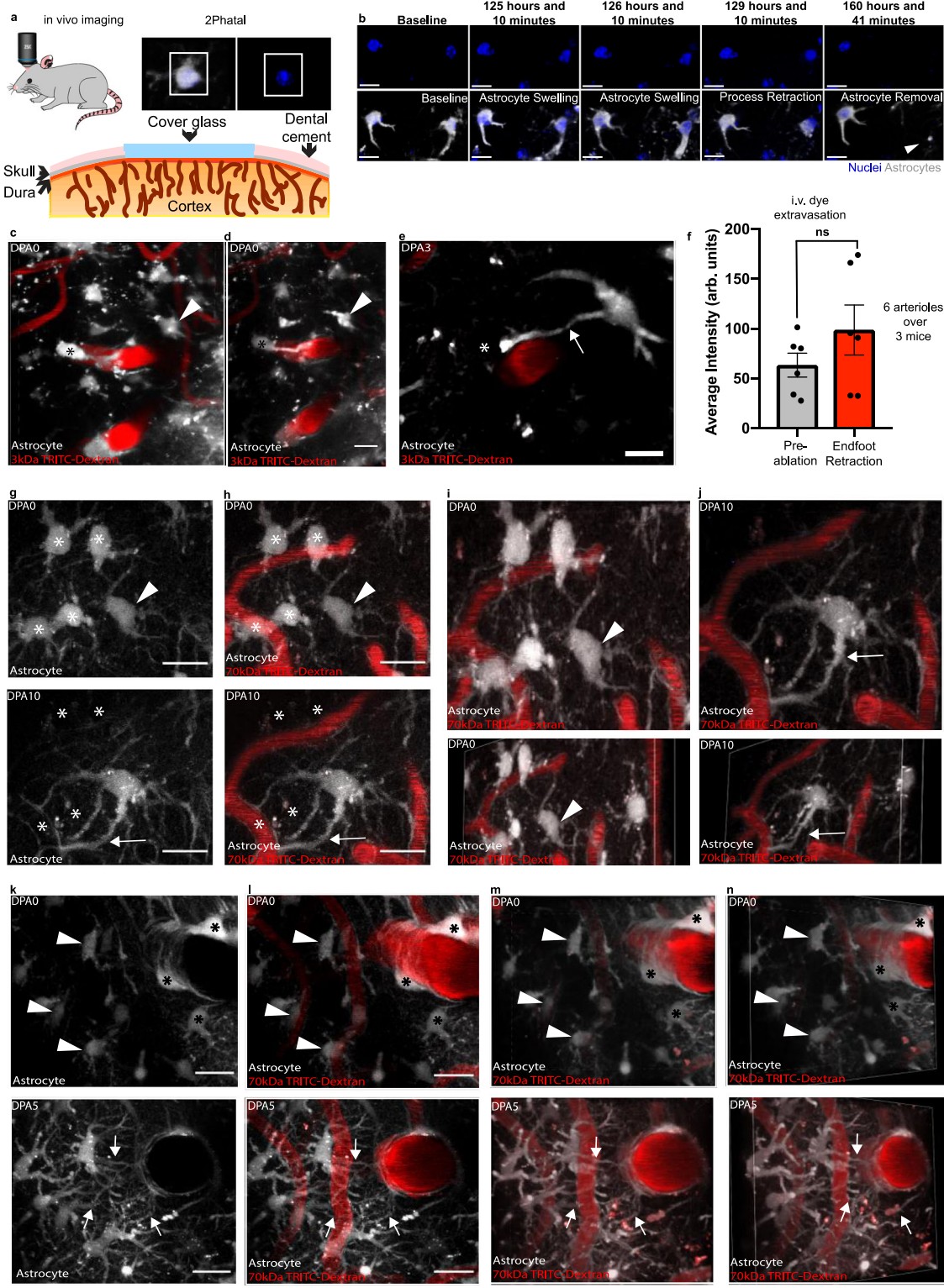

reduced relative to baseline (Supplementary Fig. 4h, i), with averaged VAA post-ablation values being 45% of baseline values. In contrast, averaged PAA post-ablation values were 95% of baseline values, showing a near complete replacement (Supplementary Fig. 4j). Taken together, these data suggest that focal loss of single astrocytes is sufficient to induce an endfoot plasticity response from nearby surrounding astrocytes with the degree of endfoot restoration dependent upon the type of astrocyte lost (VAA vs. PAA).

**Replacement endfeet can vasoconstrict precapillary arterioles**. Astrocytes have been reported to mediate neurovascular coupling at precapillary arterioles[9]. Furthermore, laser-activation of astrocytes has been shown to be a convenient way to probe their contribution to vascular physiology, as it induces a focal rise in intracellular calcium which subsequently leads to the release of vasoactive molecules[39]. In order to determine if replacement endfeet have the machinery to perform neurovascular coupling, and thus cause changes in blood vessel diameter, we ablated

**Fig. 1 The focal ablation of an astrocyte induces a gliovascular structural plasticity response at all levels of the vascular tree.** To determine if the ablation of an astrocyte(s) is sufficient to induce breaches in blood-brain barrier integrity, we utilized the in vivo single-cell 2Phatal cellular ablation method. In all images, asterisks indicate the ablated astrocyte(s) (shown in gray, blood vessels are in red), arrow heads the replacement astrocyte to-be, and arrows indicate replacement processes. **a** Cartoon diagram depicting the experimental approach. By using low laser-power to activate Hoechst (blue), we could **b** visualize astrocytes in Aldh1l1-eGFP mice up to and beyond removal of their cell body and associated processes, scale bar = 15 μm. **c** Volumetric reconstruction of an astrocyte at a penetrating arteriole on day post-ablation 0 (dpa0), with **d** maximum intensity projection of the same field, scale bar = 10 μm, and **e** maximum intensity projection at dpa5 showing no apparent disruption to blood-brain barrier integrity. Instead, an astrocyte extending a process to the vacant vascular location can be seen, scale bar = 10 μm. **f** Average intensity quantification of 3 kDa TRITC extravasation at baseline relative to the moment of endfoot retraction. $n = 6$ vessels across 3 mice, two-tailed paired end t-test, $p < 0.1100$. **g** and **h** Maximum intensity projection of astrocytes surrounding capillaries at dpa0 (top) and dpa10 (bottom), scale bar = 15 μm. **i** Volumetric reconstruction at dpa0 of capillary field showing both dorsal (top) and dorsolateral (bottom) views compared to **j** at dpa10. The arrow indicates replacement processes from neighboring astrocytes. Representative image for $n = 40$ astrocytes across 4 mice. **k** and **l** Maximum intensity projection of an ascending venule at dpa0 (top) and dpa5 (bottom). Arrows represent replacement processes from neighboring astrocytes, scale bar = 20 μm. **m** Volumetric reconstruction showing dorsal view at dpa0 (top) and dpa5 (bottom). **n** Same image from the dorsolateral view at dpa0 (top) and dpa5 (bottom). Representative image for $n = 40$ astrocytes across 7 mice. Data are presented as mean values ± SEM. LUTs have been adjusted to emphasize the replacement astrocytes and associated processes.

astrocytes in Aldh1l1-cre x GCaMP5G mice occupying vascular territories on precapillary arterioles. We then laser-activated replacement astrocytes that extended processes to the vacant vascular locations (Fig. 2a, b). We additionally activated astrocytes originally occupying vascular territory (Fig. 2c), or those astrocytes that appeared to lack any association with the vasculature via genetically encoded fluorescent labeling (Fig. 2d). This method was chosen as it allows for the direct stimulation of a specific astrocyte, and thus, any observed vessel responses following stimulation can be attributed to that stimulated astrocyte. With this methodology, we consistently observed the laser stimulation triggering an increase in intracellular calcium, immediately followed by a decrease in vessel diameter for both replacement astrocytes (Fig. 2e–h, l, and Supplementary Video 1) and original astrocytes (Fig. 2h–l and Supplementary Video 2). Compared to original astrocytes, the induced constriction by replacement astrocytes occurred with a similar, albeit slightly faster, kinetic profile (Fig. 2l). Neither the maximal calcium or vessel response were significantly different between the two (Fig. 2p). Taken together, the aforementioned data shows that replacement astrocytes can induce a maximal vessel response similar to original astrocytes upon laser-stimulation.

As mentioned above, we also laser-stimulated astrocytes lacking any apparent morphological association with the vasculature (Fig. 2d). Despite the laser-induced calcium rise being the highest in these negative control astrocytes, we observed neither a visual or measurable vessel response upon stimulation (Fig. 2h, l, m–p). Taken together, these results demonstrate that vascular interaction by astrocyte processes is necessary to exert a vessel response by laser-stimulation, and that replacement astrocytes can induce a maximal vessel response similar to original astrocytes upon laser-stimulation.

**Endfoot replacement slows with aging.** Studies in humans[40], rodents[41–43], and primates[44,45] indicate that astrocytic morphology in aged astrocytes differs markedly from young astrocytes, and other reports have shown that endfeet actually retract later in life[21]. We, therefore, asked how aging would impact focal endfoot replacement. First, we sought to confirm that this process remained intact at all levels of the vascular tree, consistent with the observations in young mice. 2Phatal ablation of several astrocytes at arterioles, venules, and capillaries revealed that this was indeed the case (Fig. 3a–f). Next, to determine if aging impacted the fidelity of replacement, we quantified the number of cells extending or growing new processes to the vacant vascular region at each vessel type, and found no difference between age groups (Fig. 3g–i).

A recent study documented an enhanced and prolonged astrogliosis response in aged mice following traumatic brain injury[46]. This suggests that aging could impact the time course of an astrocyte response, rather than just the extent of it. We therefore sought to determine if the kinetics of replacement significantly slowed with aging. Engaging in long-term repetitive in-vivo imaging revealed that this was indeed the case. 2–4-month-old mice, on average, had a replacement endfoot in place 17 min prior to complete endfoot retraction of the ablated cell (Fig. 4a–c). In contrast, however, this process was significantly slower in aged mice- occurring on average 112 min following endfoot retraction of the ablated cell (Fig. 4a–d, analysis in Fig. 4e). The time required for endfoot replacement and corpse removal post-ablation was significantly longer in aging as well (Fig. 4f, g). The area of eGFP+ endfoot covering of the vessel post-ablation showed a small but significant reduction relative to baseline (Fig. 4h, i), as was also the case in young mice (Supplementary Fig. 4i, j).

Finally, given that aged animals had vascular vacancies unoccupied for roughly two hours following endfoot retraction of the previously ablated astrocyte, we again aimed to determine if 3 kDa TRITC would extravasate at this location. Our results revealed this to not be the case (Supplementary Fig. 5) and suggests that vascular vacancies unoccupied by astrocytes for this duration of time are not sufficient to disrupt BBB integrity.

**Pharmacological inhibition of STAT3 phosphorylation via subcutaneous injection of AG490 significantly impairs the endfoot plasticity response.** When first questioning which signaling pathways might underlie gliovascular structural plasticity, we noted that the phenotype observed in Fig. 1 at venules and capillaries appeared markedly like reactive astrogliosis. This suggested that molecules previously shown to be mediators of gliosis would be valid candidates to explore. The phosphorylation of the signal transducer and activator of transcription 3 (pSTAT3) molecule by janus kinase 2 (JAK2) has long been known to underlie astrogliosis, as its pharmacological and/or genetic inhibition results in a significantly dampened gliotic response. This is evidenced by a significant reduction in glial fibrillary acidic protein (GFAP), which is considered to be a marker of reactive astrogliosis[47,48]. To test the hypothesis that pSTAT3 is necessary for focal endfoot replacement, we first subcutaneously injected the JAK2 inhibitor AG490 over a seven-day period (Fig. 5a). Comparing the volume of eGFP signal in astrocyte processes post-ablation in AG490 to vehicle-injected control animals revealed a significantly attenuated response (Fig. 5b–f), suggesting that pSTAT3 is indeed a necessary arbiter of the focal endfoot replacement response. We further wanted to assess if any perturbations in BBB integrity ensued at a penetrating arteriole after

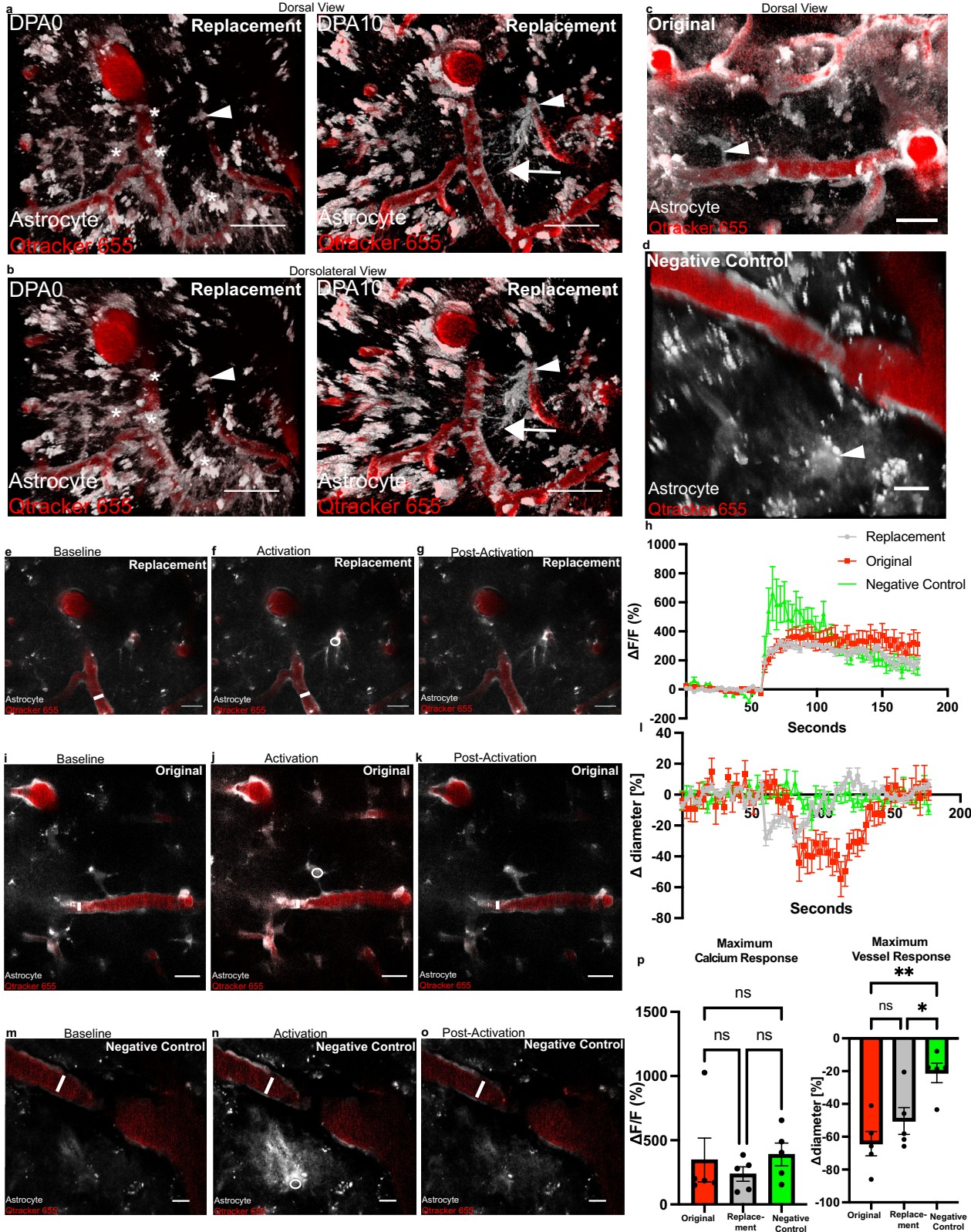

an attenuated plasticity response, and as our previous results would suggest, this was not the case (Fig. 5g–h).

Given that we only attenuated an overall increase in astrocyte volume following one dosage per day of AG490, we wanted to determine how BBB integrity might be impacted if we significantly reduced overall astrocyte volume at arterioles post-ablation while increasing AG490 dosage to three times per day. Results revealed

that, though we were able to significantly reduce the gliovascular structural plasticity response, we were not able to completely abolish it. Furthermore, even in locations along penetrating arterioles that were severely stripped of endfoot coverage, the BBB remained intact (Supplementary Fig. 6b–j). Taken together, these results suggest that a significant loss of endfoot coverage is not sufficient to disrupt BBB integrity.

**Fig. 2 Replacement endfeet vasoconstrict precapillary arterioles.** To determine if replacement endfeet can vasoconstrict precapillary arterioles, we 2Phatal ablated astrocytes in Aldh1l1cre x GCaMP5G mice, targeting astrocytes at the first branching capillary segment from an Alexa-633 hydrazide positive penetrating arteriole. Annotation is as follows: asterisks indicate ablated astrocytes (shown in gray, blood vessels are in red), arrows indicate replacement astrocyte processes, and arrow-heads indicate replacement astrocyte to-be. **a** Volumetric reconstruction showing a dorsal view of the field of interest at baseline (left) and 10 days post-ablation (right). **b** Dorsolateral view of same field. Volumetric reconstruction depicting a field of interest, with the arrow-head indicating an original astrocyte chosen for laser-activation, as in (**c**), or an astrocyte not interacting with the vasculature (negative control), as in (**d**). **e–g** Single optical section of the replacement astrocyte field at baseline, activation, and return to baseline post-activation, respectively. **h** Change in fluorescence over baseline fluorescence ($\Delta f/f$) for the duration of the experiment. **i–k** Single optical section of the original astrocyte field at baseline, activation, and return to baseline post-activation, respectively. **l** Change in vessel diameter over baseline diameter for the duration of the experiment. **m–o** Single optical section of the negative control astrocyte field at baseline, activation, and return to baseline post-activation, respectively. **p** Quantifications of maximum change in fluorescence over baseline fluorescence ($\Delta f/f$, calcium response) (left) and maximum change in diameter (right) for the duration of the experiment. Astrocytes originally occupying an appositional vascular location = red, replacement astrocytes= gray, astrocytes not interacting with the vasculature= negative control, green. $n = 5$ astrocytes over 3 mice for all groups. A Kruskal–Wallis one-way ANOVA test was performed for the $\Delta f/f$ analysis, followed by a Dunn's multiple comparison test. Data is not significantly different for all comparisons. A one-way ANOVA was performed for the maximal diameter analysis, followed by Šídák's multiple comparison test. Original vs. replacement-$p < 0.4888$, original vs. negative control-$p < 0.0036$, replacement vs. negative control-$p < 0.0421$. Scale bars = 20 μm. Data are presented as mean values ± SEM. LUTs have been adjusted to emphasize the replacement astrocytes and associated processes.

Microglial cells are known to rapidly repair the BBB following acute vessel injury[49], and their presence at the vasculature could be an additional explanation underlying maintained BBB integrity following astrocyte ablation. To examine this possibility, we used 2Phatal ablation on sulforhodamine-101 (SR101)-labeled astrocytes in CX3CR1-eGFP+ mice, which display fluorescently tagged microglial cells with eGFP under control of the CX3CR1 promoter. Longitudinal imaging in these mice showed that microglial cells do become activated two to three days following astrocyte ablation, and are present at the blood vessel to engulf 100% of ablated astrocytes (Supplementary Fig. 7a, b, f). Microglia then remained at that appositional vascular location for the duration of the experiment (Supplementary Fig. 7c).

The P2RY12 receptor is known to be necessary for microglial cell migration in response to BBB injury[49]. In light of this, we repeated the previous experiment in P2RY12 knockout x CX3CR1-eGFP mice. Interestingly, we observed no reductions in the frequency of eGFP+ microglial cell engulfment of dying astrocytes following ablation (Supplementary Fig. 7d–f). This suggests that purine release from dying cells does not autonomously regulate microglial phagocytosis of dying cells. One study demonstrated that the receptor tyrosine kinase *Mertk* regulates the velocity of microglial engagement with dying neurons[50], so it is possible that such impairments exist in P2RY12 KO animals. Finally, in order to determine if microglia are necessary to induce the replacement response, we utilized PLX3397 to pharmacologically ablate microglia in CX3CR1-eGFP+ mice following our previously published protocol with some slight modifications[51] (Supplementary Fig. 7g). Ablating astrocytes following microglial depletion revealed that endfoot replacement still occurred 100% of the time (Supplementary Fig. 7h–j). Taken together, this data shows that although microglial activation occurs following astrocyte ablation, the microglial activation is not necessary for an astrocyte plasticity response to occur. Observations of microglial activation could possibly be one reason as to why we fail to see breakdown of the BBB in locations devoid of endfoot coverage. The fact that endfoot plasticity remains intact following the pharmacological ablation of microglia, however, suggests that the BBB would remain intact following astrocyte ablation without microglia, simply due to the rapid replacement response by surrounding astrocytes.

**Genetic ablation of STAT3 phosphorylation significantly impairs the endfoot plasticity response.** Given that AG490 injection would inhibit STAT3 phosphorylation in multiple cell types, we wanted to determine if inhibition of STAT3

phosphorylation in astrocytes alone is sufficient to attenuate the endfoot plasticity response. To answer this question, we utilized Ald1h1cre-ERT2 x pSTAT3 fl/fl mice, which have loxP sites flanking exon 22 of the STAT3 gene. This region encodes a tyrosine residue (Tyr705) critical for STAT3 activation, and upon Cre activation, this tyrosine residue is excised, resulting in impairment of STAT3 activation[47]. To induce Cre activity, tamoxifen was injected once per day for five consecutive days. Cranial windows were implanted two weeks after the last day of tamoxifen injection, with commencement of imaging experiments one week after implantation (Fig. 6a).

We first aimed to recapitulate the results obtained from AG490 studies. Specifically, we ablated SR101-labeled VAA astrocytes and measured the change in volume of individual replacement astrocytes from baseline. We chose VAA astrocytes as we thought it would elicit the largest response, and performed the same number of ablations across groups. Results revealed that astrocytes in control Cre− mice exhibited, on average, a 250% increase from baseline five days post-ablation, with clear replacement processes at the vacant vascular region (Fig. 6b, top and bottom image, quantified in d). In contrast to this, astrocytes in Cre+ experimental mice exhibited a 150% increase in volume from baseline without apparent processes at the vacant vascular location five days post-ablation (Fig. 6c, top and bottom image, quantified in d). We next quantified the area of eGFP+ endfoot lining around the FITC-Dextran labeled vasculature pre-and post-ablation in response to PAA ablation. Replacement astrocyte processes were visible at the vacant vascular location in Cre− mice (Fig. 6e, top and bottom image), and the post-ablation area of endfoot lining remained consistent relative to baseline (Fig. 6g). In contrast, experimental Cre+ mice exemplified a clear reduction in replacement post-ablation (Fig. 6f, top and bottom image) and the area of eGFP+ endfoot lining was reduced, at just under 50% on average (45% area relative to baseline) (Fig. 6g). Taken together, these results show that pSTAT3 activation in astrocytes is indeed necessary to achieve endfoot replacement at arterioles in response to PAA ablation.

**2Phatal as a model of focal endfoot replacement following astrocyte loss post- transient photothrombotic stroke.** We initially turned to 2Phatal to model a loss of endfoot coverage on the underlying vasculature based off what we had previously characterized in two disease conditions[24,25]; however, 2Phatal results in the complete loss of an astrocyte cell body and associated processes as opposed to just an endfoot. We, therefore,

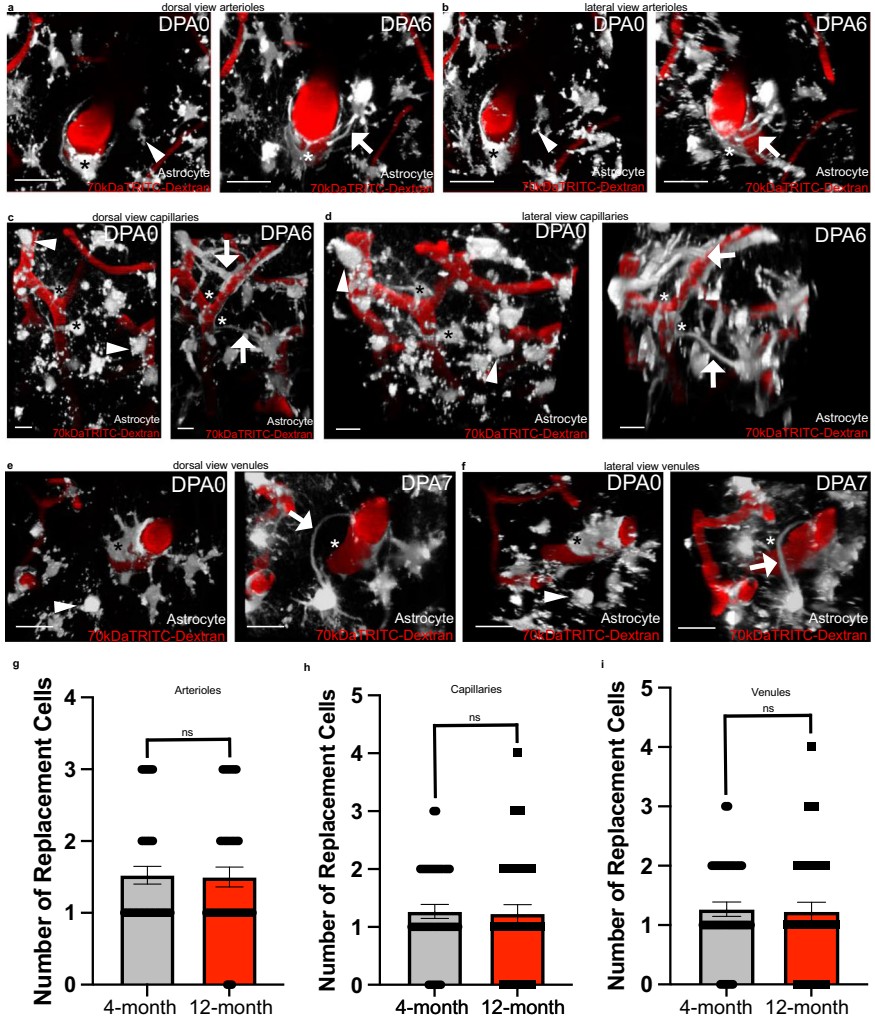

**Fig. 3 Gliovascular structural plasticity remains intact at all levels of the vascular tree in aging.** In order to determine if gliovascular structural plasticity remained intact in aging, we ablated astrocytes in 12-month-old Aldh1l1-eGFP mice making contact with Alexa 633 hydrazide positive penetrating arterioles, capillaries, and venules. We further compared the extent of replacement in old versus young mice as well. In all images, asterisks indicate the ablated astrocyte(s) (shown in gray, blood vessels are in red), arrow heads indicate the eventual replacement astrocyte, and arrows demarcate replacement processes. Volumetric reconstruction showing astrocyte associations with a penetrating arteriole at dpa0 (left) and dpa6 (right), from **a** a dorsal perspective and **b** a dorsolateral perspective. Scale bar = 25 µm. $n = 40$ astrocytes across 6 mice. **c** and **d** Volumetric reconstruction showing astrocytes and their association with a capillary at dpa0 (left) and dpa6 (right) from **c** a dorsal view and **d** a dorsolateral view. Scale bar = 10 µm. $n = 40$ astrocytes over 4 mice. Volumetric reconstruction showing an ascending venule and astrocyte interactions with it at dpa0 (left) and dpa7 (right), from **e** a dorsal view and **f** a dorsolateral view. Scale bar = 20 µm. $n = 40$ astrocytes across 5 mice. **g** Average number of replacement astrocytes (the number of cells extending processes to vascular vacancies) in 4- versus 12-month old mice at arterioles, Two-tailed Mann–Whitney test, $p = 0.9465$, $n = 40$ cells examined over 7 mice for 4-month data, $n = 40$ cells examined over 6 mice for 12-month data. **h** Average number of replacement astrocytes in 4-versus 12-month old mice at capillaries, Two-tailed Mann–Whitney test, $p = 0.6853$, $n = 40$ cells examined over 4 mice for both 4- and 12-month old mice. **i** Average number of replacement astrocytes in 4-versus 12-month old mice at venules, Two-tailed Mann–Whitney test, $p = 0.3060$, $n = 40$ cells examined over 7 mice for 4-month old data, $n = 40$ cells examined over 5 mice for 12-month old data. Data are presented as mean values ± SEM. LUTs have been adjusted to emphasize the replacement astrocytes and associated processes.

wanted to determine if this 2Phatal ablation procedure might more closely model other disease conditions. Given that 2Phatal presumably triggers apoptosis through ROS-induced DNA damage[31], we searched for reports on any disease conditions marked by astrocyte cell death at the vascular interface due to ROS-induced DNA damage, finding one such candidate in ischemia-reperfusion (I-R), a condition wherein blood-flow is restored to a vessel following an ischemic insult due to vessel occlusion. Others have focused already on how I-R impacts neuronal health; however, astrocytes have been reported to be just as sensitive as neurons to reperfusion-induced apoptosis following the restoration of blood flow[52].

We, therefore, set out to model this condition in vivo by utilizing the Rose Bengal photothrombosis stroke model. Rose Bengal is a light-sensitive dye that, upon encountering a green laser, undergoes nucleation and forms a clot. Successful clot formation can be visualized by a dark area forming in the vessel, indicative of red blood cell accumulation, and intense fluorescence due to dye accumulation above that dark mass (Fig. 7d–f)[53]. Critically, this method allowed us to focally and transiently occlude penetrating arterioles and thereby examine if a structural plasticity response would occur following focal loss of astrocyte-vascular coverage. Following reperfusion, we searched regions below the plane of dye nucleation for signs of astrocyte cell death

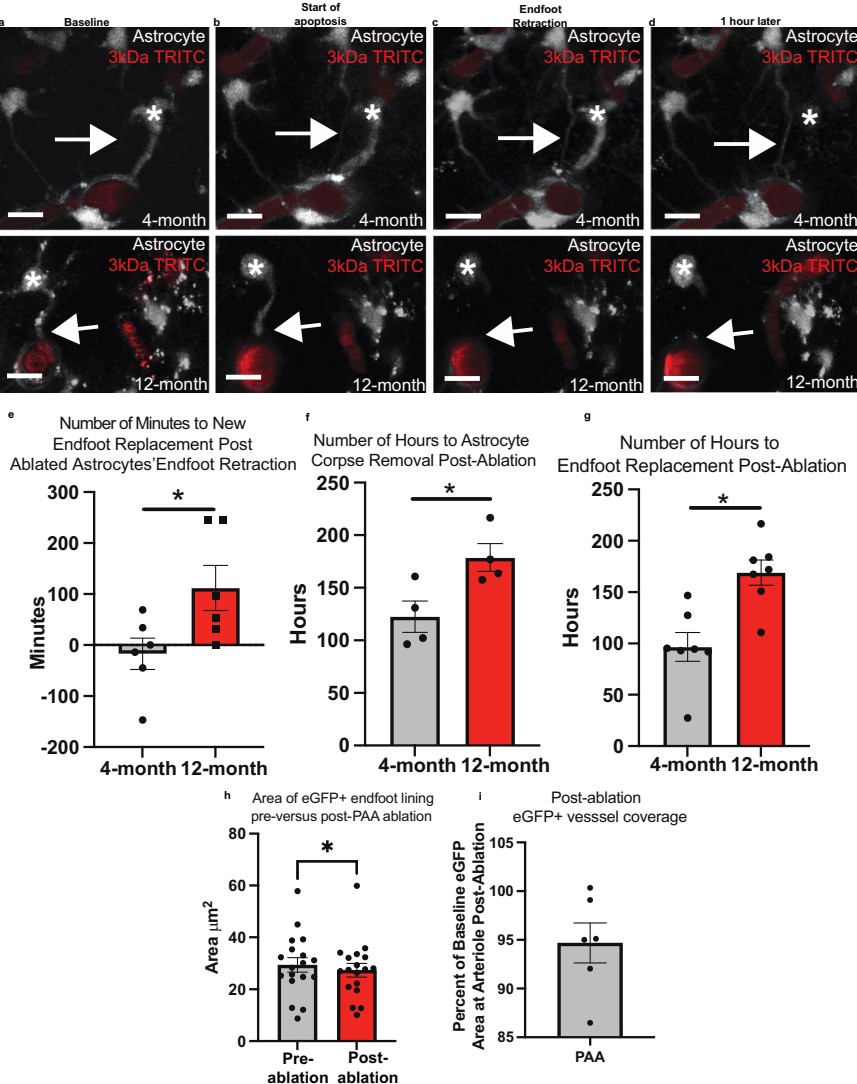

**Fig. 4 Aging significantly attenuates the kinetics of the gliovascular structural plasticity response.** In order to determine if the kinetics of endfoot replacement significantly slowed as a result of aging, we ablated astrocytes at penetrating arterioles and acquired continual z-stacks to capture the exact time of endfoot replacement. White asterisks indicate ablated astrocytes (shown in gray, blood vessels are in red) and arrows demarcate replacement processes (as in **a** and **b**) or lack thereof (as in **c** and **d**). Volumetric reconstruction showing a penetrating arteriole in a 4-month-old mouse (top) and 12-month old mouse (bottom) at **a** baseline, **b** the start of apoptosis, **c** the moment of endfoot retraction, and **d** one hour later. Scale bar = 10 μm. $n = 6$ astrocytes across 4 mice. **e** Average number of minutes to new endfoot replacement post-ablation in 4- versus 12-month old mice. Two-tailed, unpaired t-test, $p < 0.0371$. **f** Average number of hours to astrocyte corpse removal post-ablation in 4- versus 12-month old mice. Two-tailed, unpaired t-test, $p < 0.03$. **g** Average number of hours to endfoot replacement in 4- versus 12-month-old mice post-ablation. Two-tailed, unpaired t-test, $p < 0.0022$. $n = 6$ cells across 4 mice for both age groups. **h** Average area of eGFP+ endfoot lining around arterioles at ablated location pre- versus post-ablation. Two-tailed paired end t-test, $p < 0.0370$, $n = 18$ optical sections/4 mice. **i** Average post-ablation eGFP+ endfoot vessel lining area expressed as a percentage of baseline. $n =$ Focal PAA ablations at 6 arterioles/4 mice. Note that the data points in **h** are averaged to give one dot in (**i**). Data are presented as mean values ± SEM. LUTS have been adjusted to emphasize the replacement astrocytes and associated processes.

at the vascular interface, using the surrounding vascular profile to ensure that we were comparing the same regions pre-and post-vessel occlusion (Fig. 7g, h). Cell death was indeed observed, and in the days following, surrounding astrocytes reached out to that vacant vascular location (Fig. 7i, j, compare to baseline images in Fig. 7a, b). This data suggests that the 2Phatal ablation of astrocytes at vascular interfaces could potentially be thought of as a model for focal loss of endfoot coverage due to reperfusion-induced apoptosis. Given that stroke does lead to neural injury and subsequent astrogliosis, these data further support the notion that gliovascular plasticity is a focal gliosis response aimed at ensuring continual vascular coverage by astrocytes.

## Discussion

Previous studies suggest that a number of nervous system insults and diseases present with impaired gliovascular interactions and even BBB disruption. Here we set out to determine if focal ablation of single astrocytes is sufficient to compromise BBB integrity at the site of cellular ablation. We had previously shown that focal endfoot separation due to invading glioma cells resulted in extravasation of various molecular weight dextran dyes and significant losses in tight junction proteins zonula-occludens-1 and Claudin 5[24]. By employing the 2Phatal ablation technique to induce single-cell apoptosis, we found that focal loss of an astrocyte did not in fact compromise BBB integrity (Fig. 1e, f), but

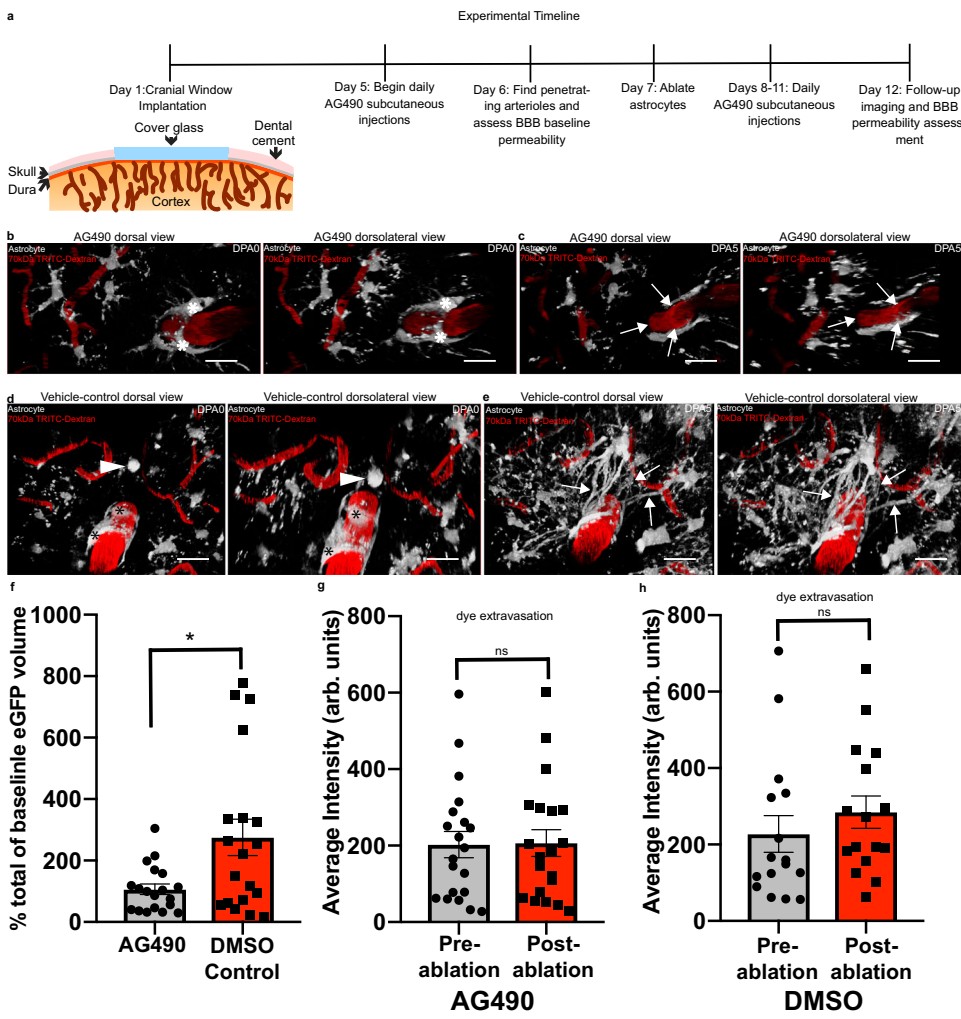

**Fig. 5 The pharmacological inhibition of EGFR/pSTAT3 significantly reduces the volume of replacement astrocytes post-ablation.** In order to determine if EGFR and pSTAT3 are necessary for gliovascular structural plasticity, we subcutaneously injected AG490. In all images, asterisks indicate ablated astrocyte(s) (shown in gray, blood vessels are in red), arrow heads indicate the eventual replacement astrocyte, and arrows demarcate replacement processes, or lack thereof. **a** Schematic illustrating the experimental timeline. Volumetric reconstructions showing the dorsal view (left) and dorsolateral view (right) of astrocytes and their interactions with a penetrating arteriole in AG490-injected mice at **b** dpa0 and **c** dpa5. Scale bar = 20 µm. Volumetric reconstructions showing the dorsal view (left) and dorsolateral view (right) of astrocytes interacting with a penetrating arteriole in DMSO vehicle-injected control mice at **d** dpa0 and **e** dpa5. Scale bar = 20 µm. n = 19 fields of view over 4 mice. **f** Bar graph comparing percent increase in eGFP volume at dpa5 relative to baseline in AG490-injected mice relative to DMSO-injected control, n = 19 penetrating arterioles over 4 mice, Two-tailed Mann–Whitney test, p = 0.0497. **g** Bar graph comparing the average intensity of 3 kDa TRITC extravasation in AG490 injected mice pre-ablation versus post-ablation, n = 20 penetrating arterioles over 4 mice, Two-tailed Wilcoxin matched pairs signed-rank test, p = 0.6215. **h** DMSO vehicle-injected control group, n = 16 penetrating arteriole over 4 mice, Two-tailed Wilcoxin matched pairs signed-rank test, p = 0.1167. Data are presented as mean values ± SEM. LUTS have been adjusted to emphasize the replacement astrocytes and associated processes.

instead reliably induced a plastic response whereby surrounding astrocytes reach out their processes to fill the vascular vacancy left by the ablated astrocyte (Fig. 1e). Moreover, the BBB remained intact in conditions where the vasculature was vacant for up to two hours (Supplementary Fig. 5c) or was almost entirely stripped of endfoot coverage (Supplementary Fig. 6).

These results are interesting given a recent study[32] demonstrating the necessity of astrocytes in maintaining BBB integrity. These prior findings were obtained using a sparser and more permanent astrocyte ablation without evidence of endfoot plasticity; furthermore, they relied on extravasation of ~1 kDa Cadaverine as a marker for BBB disruption. Unfortunately, in our studies we found this dye flawed in its ability to discriminate between normal and abnormal BBB function, since we observed baseline leakage in control mice (Supplementary Fig. 3a). It is possible that ablating greater numbers of astrocytes per region,

as opposed to our one-to-one replacement model, would result in a reduced or completely abolished endfoot plasticity response. A future study may seek to elucidate the potential threshold at which loss of astrocytes- on a much larger scale- could fail to trigger the plasticity response and lead to loss of BBB integrity. Another recent study[54] that used albumin, a reliable extravasation marker, showed that an astrocyte-specific connexin-30 and -43 double knockout resulted in both swollen astrocytic endfeet and impaired BBB integrity. However, this was most prevalent in deep brain structures such as the striatum and basal ganglia rather than cortical regions, which is very similar to what was demonstrated upon deletion of astrocyte-specific laminins[55]. Note that care must be taken to account for changes in pericyte support, given that pericyte-deficient mouse models have been shown to alter astrocyte properties[56]; therefore, loss of astrocytes may have affected pericyte coverage and

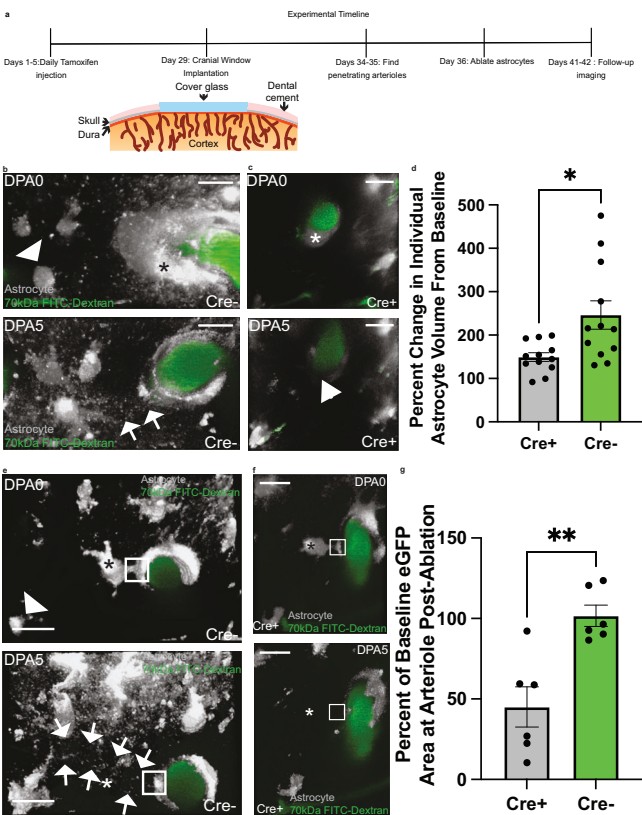

**Fig. 6 The genetic ablation of pSTAT3 significantly reduces the volume of replacement astrocytes and eGFP+ area around the vasculature post-ablation.** In all images, asterisks indicate the ablated astrocyte(s) (shown in gray, blood vessels are in green) and arrows demarcate replacement processes. Arrowheads indicate which astrocytes at baseline will serve as replacement astrocytes post-ablation. **a** Cartoon diagram depicting the experimental approach. **b** Volumetric reconstruction of an astrocyte at a penetrating arteriole in control Cre− mice on (top) day post-ablation 0 (dpa0) and (bottom) day post-ablation five (dpa5), in which a surrounding astrocyte replaces the vacancy left by the previously ablated vessel-associated astrocyte. **c** Volumetric reconstruction of an astrocyte at a penetrating arteriole in experimental Cre+ mice on (top) dpa0 and (bottom) dpa5, in which surrounding astrocytes fail to replace the vacancy left by the previously ablated vessel-associated astrocyte. The box in the bottom image highlights the vacant vascular region. $n = 12$ astrocytes across 3 mice. **d** Percent change in individual astrocyte volume from baseline. $n = 12$ astrocytes across 3 mice for both groups, Two-tailed Unpaired t-test with Welch's correction, $p < 0.0144$. **e** Volumetric reconstruction from a dorsolateral view of an astrocyte at a penetrating arteriole in control Cre− mice on (top) dpa0 and (bottom) dpa5, in which a surrounding astrocyte replaces the vacancy left by the previously ablated parenchymal-associated astrocyte. eGFP area around the vessel remains consistent as highlighted in the boxed region. **f** Volumetric reconstruction of an astrocyte at a penetrating arteriole in experimental Cre+ mice on (top) dpa0 and (bottom) dpa5, in which surrounding astrocytes fail to replace the vacancy left by the previously ablated parenchymal-associated astrocyte. The eGFP area around the vessel is significantly reduced at dpa5, as highlighted by the boxed region. $n = 6$ astrocytes over 3 mice for both groups. **g** Percent change in eGFP area around the vessel relative to baseline. $n = 6$ astrocytes over 3 mice for both groups, Two-tailed Unpaired t-test with Welch's correction, $p < 0.0043$. Scale bar is 20 μm in all images. Data are presented as mean values ± SEM. LUTs have been adjusted to emphasize the replacement astrocytes and associated processes.

thus indirectly altered BBB integrity. To avoid a confounding contribution of pericyte dysfunction to BBB integrity, we exclusively studied penetrating arterioles lacking pericyte presence.

While it is conceivable that the role of astrocytes in maintaining BBB integrity may be region-specific, which aligns with evidence supporting astrocyte functional heterogeneity in the brain[57], loss of BBB integrity might also depend on the duration of the loss of astrocyte coverage. We believe that our data points to the rapid plasticity or repair response by neighboring astrocytes as the primary reason that vessel function and BBB integrity are unaffected by the loss of a single or few astrocytes. This is in excellent agreement with a recent study that focally ablated pericytes, which similarly did not damage BBB integrity, but did result in a comparable plasticity response[26]. Yet, global pericyte-deficient mouse models have been clearly shown to perturb the BBB[56]. Unlike focal pericyte ablation, which resulted in an absence of pericyte-capillary coverage for days, focal ablation of astrocytes in our hands results only in a lapse of endfoot coverage for minutes to hours. Indeed, we were surprised to find that the replacement processes were already in place ahead of complete retraction of the lesioned cell by a few minutes. Given that the half-life for the tight junction protein ZO-1 is 5.2 h in MDCK cells[58], and 90 min for claudin-5[59], it is likely that we were unable to strip a vessel of endfoot contact long enough to breach the barrier- assuming focal ablation is sufficient to do so. It is further possible that the pharmacological inhibition of STAT3 prevented BBB breakdown following astrocyte ablation, as other studies have documented restoration of BBB integrity following prevention of STAT3 activation via inhibition of JAK[60]. Based on our results, we hypothesize that if enough vascular territory is stripped for a long enough period of time, BBB leakage will be induced. The speed of the plasticity response in our studies prevented this observation from being made; in other words, the kinetics of the response seems to protect and retain the integrity of the BBB. Future studies focally ablating astrocytes should aim to do so in conditions where the replacement of endfeet is either stalled for longer periods, or is completely abolished. Alternatively, such studies may find—as was the case in focal pericyte ablation studies- that a more permanent focal ablation of astrocytes is not sufficient to perturb BBB integrity. It should also be considered that astrocyte roles may differ with the nature, severity, and extent of an insult, particularly regarding the difference between roles in (a) contributing to an initial BBB breakdown, versus (b) repairing or restoring a comprehensive loss of BBB function triggered by severe acute tissue damage. For example, naturally occurring repair of the BBB after its complete local disruption by traumatic injury or irreversible stroke is dependent on astrocytes—such that experimental ablation of astrocytes prevents BBB repair, and grafting of replacement astrocytes can restore it[61,62].

Beyond the BBB, we also built off a previous study that documented endfoot plasticity[63] and further determined that it occurs at all levels of the vascular tree (Fig. 1g–n). Furthermore, this process begins even prior to the ablated astrocyte having its corpse entirely removed in young mice, whereas aging significantly slows down the swift kinetics of replacement (Fig. 4a–f). These results are interesting given our observations of endfoot plasticity to vascular vacancies left by dying astrocytes post-focal photothrombotic stroke (Fig. 7). It is known that the probability of stroke occurrence following transient ischemic attack increases with aging[64], and presumably any reduction of, or lapse in, endfoot coverage could affect vessel recovery following ischemic onset—potentially contributing to this increased stroke probability. Our results would suggest that there would be a lapse in endfoot coverage with aging only as endfoot coverage in response

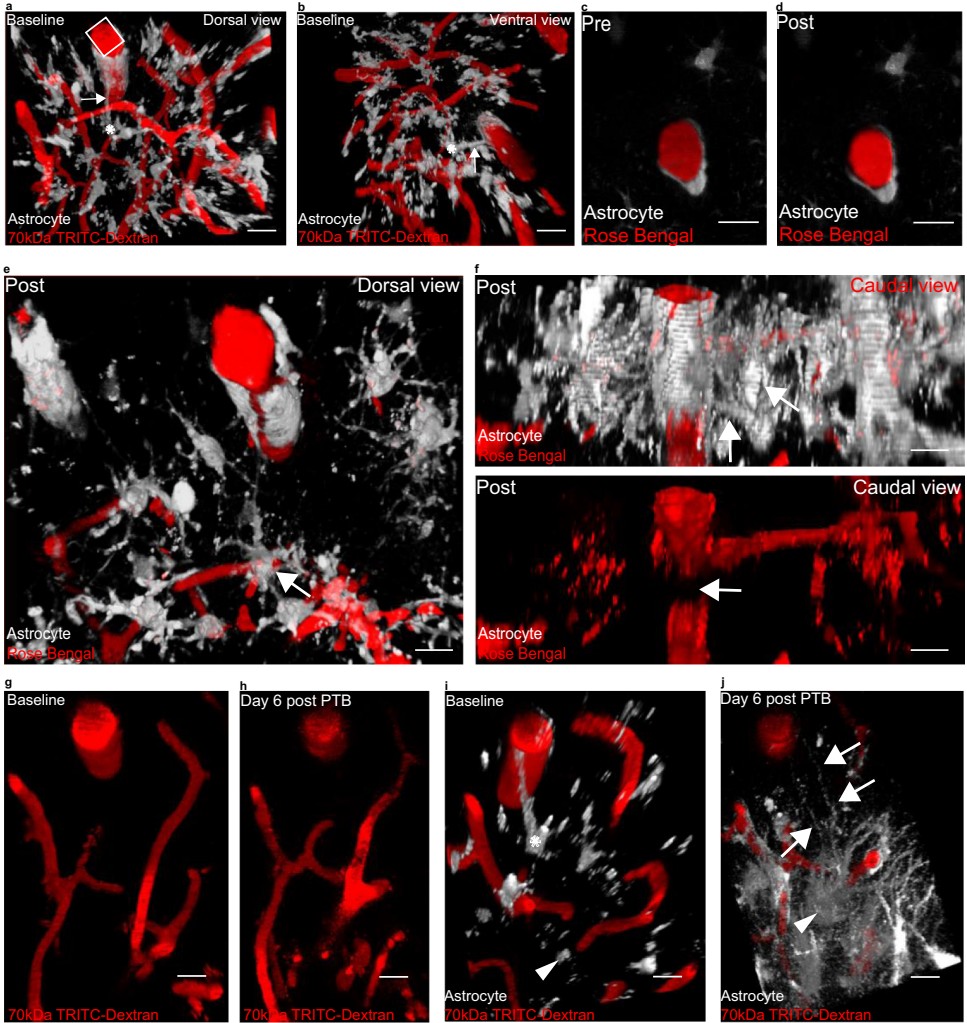

**Fig. 7 Gliovascular structural plasticity occurs following reperfusion post-focal photothrombotic stroke.** In order to determine if gliovascular structural plasticity occurs following loss of astrocyte-vascular coverage due to CNS insult, we turned to the Rose Bengal focal photothrombotic stroke method. Arrows in all images demarcate astrocyte processes (shown in gray, blood vessels are in red) either at baseline (as in **a**–**b**) or replacement processes six days following reperfusion from focal-stroke (as in **j**). Asterisks indicates an astrocyte that dies following reperfusion post-focal photothrombotic stroke. **a** and **b** Volumetric reconstruction showing a penetrating arteriole at baseline from **a** dorsal view and **b** a ventral view. Single optical section showing a penetrating arteriole **c** following Rose Bengal injection and **d** immediately after laser-induced dye nucleation. **e** Volumetric reconstruction showing a penetrating arteriole immediately after Rose Bengal laser-induced dye nucleation from a dorsal view. **f** Lateral view of same field in e) showing the penetrating arteriole (top) after Rose Bengal injection and (bottom) immediately after dye nucleation. The arrow in the bottom image demarcates where red blood cell (RBC) buildup has occurred. **g** and **h** Volumetric reconstruction showing a dorsal view of the vascular profile surrounding the occluded vessel at **g** pre-occlusion (baseline) versus **h** post-occlusion (day 6 post PTB), which was used to ensure comparison of the same location pre- versus post-vessel occlusion. **i** Volumetric reconstruction of the same field from **g** from a dorsolateral view and showing eGFP-labeled astrocytes. The asterisk indicates the astrocyte that will die following reperfusion of the transiently occluded vessel, and arrow-head indicates the replacement astrocyte to-be. **j** Volumetric reconstruction of the same field from **h**) from a dorsolateral view and showing eGFP-labeled astrocytes. The arrows highlight the replacement processes emerging from the replacement astrocytes, whose soma is indicated by the arrow-head. $n = 5$ arterioles/5 mice. Scale bar is 20 μm in all images. LUTs have been adjusted to emphasize replacement astrocytes and associated processes.

to PAA ablation is ultimately restored post-ablation, just as was observed in young mice (Supplementary Fig. 4).

Finally, pharmacologically inhibiting the phosphorylation of STAT3 by JAK2 via AG490 injection revealed pSTAT3 to be an essential regulator of the focal endfoot plasticity response. Many studies have documented AG490 inhibition of JAK2-mediated phosphorylation of STAT3[48,65,66], but we confirmed that STAT3 activation, specifically in astrocytes, is necessary to achieve endfoot replacement at arterioles in response to PAA ablation (Fig. 6b–g). Taken together, these results point to focal endfoot replacement response as being dependent on genes classically associated with astrogliosis[67]. Future studies should seek to categorize the type of

gliosis the endfoot plasticity response falls under. We speculate this is a resolving reactivity response[67], but seeking to understand if there is local proliferation following ablation will be the first step in future experiments. Additionally, other molecules regulating astrocyte migration such as Aquaporin-4 should be investigated in their potential regulation of endfoot plasticity[68].

In summary, this is the first study to characterize gliovascular structural plasticity at all levels of the vascular tree; its physiological relevance to blood flow and the BBB; and its alterations in aging. These findings add to the burgeoning literature regarding the complexities of basic astrocyte biology, and the role of the astrocyte in supporting the cerebrovasculature.

## Methods

All studies were approved by the Institutional Animal Care and Use Committees of the University of Virginia and Virginia Tech, and were conducted in compliance with the National Institutes of Health's 'Guide for the Care and Use of Laboratory Animals'.

**Mice**. For all studies, both male and female mice were used. Young mice were aged 2–4 months, and aged mice were 12-months of age. All mice were housed under controlled temperature, humidity, and light (12:12 h light-dark cycle) with food and water readily available ad libitum, with no more than 5mice/cage. The following transgenic lines were used. Swiss Webster-Aldhlll-eGFP bacterial artificial chromosome transgenic mice (generated by the GENSAT;project); NG2-dsRedBAC mice (Jackson Labs number 008241) were crossed with Aldhlll-eGFP mice, and mice homozygous for both transgenes were maintained as a colony. Aldhlll-cre (Jackson Labs number 023748) mice were crossed with CAG-GCaMP5G-tdTomato (Jackson Labs number 024477) mice. Aldhlll-cre/ERT2 mice (Jackson lab number 031008) were crossed with STAT3-loxP mice (Hermann et al., 2008[47], Takeda et al., 1998). CX3CR1-eGFP and P2RY12KO mice were a generous gift from Ukpong B. Eyo.

**In vivo multiphoton imaging through a cranial window**. Cranial windows were used for all experiments, with all surgeries being performed as described previously[20,21] with slight modifications. Following induction of surgical plane anesthesia with 2–5% isoflurane, pre-operative analgesics and antibiotics were administered intraperitoneally. Following this, the hair and skin of the skull were removed, and a $3 \times 3$ mm craniectomy anterior to lambda and posterior to bregma was subsequently performed on one hemisphere. The dura was removed next, followed by the placement of a $3 \times 3$ mm #1 cover glass that was then affixed and sealed with dental cement. All mice were allowed to recover for 5–7 days before experiments commenced. For imaging, animals were placed on a Kopf stereotax with heating pad. While imaging, animals were anaesthetized (~100 beats per minute), and their vitals constantly monitored. Cerebral vessels were visualized by retro-orbital injection of 70 kDa TRITC-Dextran (ThermoFisher catalog number D1818), 3 kDa TRITC-Dextran (ThermoFisher catalog number D3307), 70 kDa FITC-Dextran in Sulforhodamine-101 experients (ThermoFisher catalog number D1822), and/or 967 Da Cadaverine Alexa Fluor 555 (ThermoFisher catalog number A-30671), and imaging was performed 100–200 μms below the surface of soma-tosensory cortex. A Chameleon Vision II (Coherent) laser tuned to 870 nm was used to excite all dyes. Optical sections were acquired using a four-channel Olympus FV1000MPE multiphoton laser scanning fluorescence microscope. Mice undergoing imaging experiments lasting longer than two hours would receive a bolus of Lactated Ringer's solution every two hours at 15 μl/g body weight, never exceeding 500 μl per injection.

For the following studies: single astrocyte ablation; negative control vasoconstriction; the effect of astrocyte ablation on pericyte volume and capillary diameter; microglial activation following astrocyte ablation in CX3CR1-eGFP and P2RY12KO mice; endfoot plasticity in PLX3397 microglial depleted mice; and Aldh1l1-creERT2 x pSTAT3 fl/fl, an Olympus Dual beam FVMPE-RS multiphoton microscope was used. A MaiTai DeepSee laser was utilized on this multiphoton. Two cooled GaAsP and two multialkali photomultiplier detectors allowed for simultaneous 4-channel multiplexing. Importantly, both multiphotons used in this study were equipped with a XLPLN25X/1.05 NA water-immersion objective (Olympus). Z projections were created using FIJI (NIH) and Nikon Imaging Software (NIS)-Elements, with optical section thickness for each z-stack acquired with the Olympus FV1000 MPE set to 2 μm, and those from the Olympus FVMPE-RS set to 1 μm.

**2Phatal Ablation**. To induce single-cell apoptosis in astrocytes, mice underwent the surgical procedure described above. Upon removal of the dura, Hoechst 33342 dye (ThermoFisher catalog number H5370) was applied topically (0.04 mg ml$^{-1}$ diluted in PBS) to the cortex of Aldh1l1-eGFP mice over ten minutes, then washed thoroughly with cold 1XPBS. To ablate, an $8 \times 8$ μm square ROI was placed over a dual eGFP- and Hoechst-positive astrocyte nuclei whose soma was either on a vessel of interest or had endfeet contacting the vasculature. To achieve photo-bleaching, pixel dwell time was set to 100 μs/pixel, laser wavelength was set to 775 nm, and the ROI underwent scanning for a duration of 20 s. A Newport Model 1919-R power meter with a silicone-based OD3 photodetector was used to determine power at the objective for all ablation experiments with the FV1000MPE multiphoton, and a range of 2.33–53.4 mW was used for all experiments. Increased power was used at focal planes of increased depths and/or decreasing Hoechst intensity, as measured in the activation ROI. eGFP was visualized with a laser wavelength of 870 nm. For ablation experiments involving Sulforhodamine-101 (SR101)-labeling of astrocytes, SR101 was retro-orbitally injected and imaging commenced 60 min later.

### Image analysis

*Replacement astrocyte kinetics*. For analysis of replacement kinetics, only astrocytes extending clear processes to the vascular interface were chosen for ablation. Due to the intrinsic heterogeneity of time to onset of astrocyte cell death following 2Phatal

ablation, the ablated region was checked twice per day following ablation, starting at dpa1, to determine if the ablated astrocyte's soma has begun to swell. This phenotype was indicative that the ablated astrocyte would undergo phagocytosis within the next 24 h. Upon implementing a z scan during each of the twice-daily checks, the ablated region was checked for visible enlargement of the soma. If the soma appeared visibly enlarged and the endfoot process had not yet retracted, a circular ROI was applied to the area of a maximum intensity projection to determine increase (quantified at approximately 130% of the area at baseline), and repetitive scanning would begin every five minutes in order to acquire a z-stack of 2 μm optical section thickness until replacement processes were visible at the vacant vascular region. A number of instances occurred in which the swelling of the astrocyte soma was observed at the end of the event, and the complete process could not be captured (values not reported in dataset). To quantify the number of minutes to endfoot replacement in 4-month and 12-month-old mice, the time of fluorescence fading in the process of the ablated astrocyte was considered time point zero. This was determined by an inability to apply the autodetect ROI feature to an astrocyte process, which was previously possible at baseline and following ablation. From there, the number of minutes until a process from a replacement astrocyte was observed to enter the vacant vascular territory was then determined from the captured z-stacks. If the replacement process made contact with the vessel prior to time point zero (i.e., before the ablated astrocyte process fluorescence faded beyond auto-detection), this was reported as negative minutes. If the replacement process made contact with the vessel after time point zero, this was reported as positive minutes. For this study, all imaging sessions were long once commenced (8–12 h).

**Replacement astrocyte number**. To determine the number of astrocytes extending processes to occupy vascular vacancies, NIS elements was used to compare the baseline z-stack to z-stacks from later time points following complete removal of ablated astrocytes; specifically, each ablated astrocyte, and the optical section(s) its soma occupied, was denoted in the baseline image. The surrounding vascular landmarks and astrocytes that were not ablated could therefore serve as fiduciary landmarks when evaluating that same field post-ablation. Any astrocyte at post-ablation timepoints that appeared to extend processes and occupy a vascular vacancy was identified in the baseline image, again using the (1) surrounding vascular profile, (2) astrocytes that were not ablated, and (3) z location of the astrocyte in question, relative to the ablated astrocyte. If at baseline, the replacement astrocyte in question did not appear to have processes interacting with the vasculature, even upon dramatically increasing the look up table, it was considered a replacement astrocyte. The total number of cells fulfilling these criteria were reported as the total number of replacement cells.

**Post-ablation eGFP+astrocyte vessel coverage area analysis**. To determine if vascular regions are occupied by the same total area of endfoot coverage following ablation of parenchymal-associated astrocytes, acquired z stacks from the baseline recording and the day of recorded astrocyte replacement were compiled. Using NIS elements software, the same number of optical sections, selected based on surrounding reference astrocytes and vascular landmarks, were taken from both timepoints. The rotating rectangle feature was then used to select the vascular region surrounded by the ablated astrocyte, keeping parameters consistent for both images. To measure endfoot coverage around the vessel in each optical section, the autodetect ROI feature was used, indicating the area of eGFP or SR101 signal. The average of those area measurements is the value reported for each individual data point in endfoot replacement graphs shown in Figs. 4i, 6g, and Supplementary Fig. 4j. See Supplementary Fig. 8 for a graphic representation of this analysis paradigm.

**Replacement astrocyte volumetric analysis and NG2+pericyte volume post-astrocyte ablation**. For eGFP volumetric analysis in AG490 and Aldh1l1creERT2 x pSTAT3 fl/fl studies of NG2+pericyte volume following single-astrocyte ablation, images were opened in NIS elements volume viewer and underwent background subtraction using the rolling ball radius feature. A median filter and binary threshold were subsequently applied. eGFP volume of individual replacement astrocytes or NG2 volume of pericytes at baseline and post-ablation was then recorded. The same number of optical sections were used for time points being directly compared. To ensure that the same region of astrocyte ablation was compared pre- and post-ablation, the rotating rectangle feature was used, allowing for area selection in an image without changing the underlying metadata. An ROI of the same size was applied to images of time points being compared to serve as a size reference for the rotating rectangle.

**Astrocyte engulfment by microglia post-astrocyte ablation**. Following 2Phatal ablation of SR101-labeled astrocytes in CX3CR1-eGFP mice, z-stacks were acquired for consecutive days until observations of eGFP+ microglia signal overlapping with SR101 astrocyte-signal were made. Consecutive acquisition of z-stacks continued daily until the SR101-signal was no longer visible, imaging at identical timepoints following retro-orbital injection. The frequency of SR101 signal overlap with eGFP+ signal was determined and reported as the frequency of microglial

engulfment of astrocytes. The same analysis was performed in the P2RY12-KO studies.

**Assessment of capillary diameter using VasoMetrics**. Maximum intensity projections with identical optical sections numbers were opened in FIJI (ImageJ) upon engagement of the VasoMetrics program developed out of Andy Shih's laboratory (see McDowell et al. 2021). This program generates multiple cross-sectional lines placed at even intervals along a user-defined length of capillary. The length and spacing of the cross-sectional lines used to indicate the axis of the capillary at baseline and the user-defined vessel length were kept the same for pre and post-ablation images. The cross-sectional lines placed along the axis of the capillary then report diameters from the full width at half maximum of the intensity profiles, where intensity profiles are generated from the dye-filled lumen of the vessel. These values are then used to generate an average diameter for the capillary. The number of cross-sectional lines and generated values were kept consistent for pre-and post-ablation images.

**Assessment of endfoot plasticity in PLX3397 microglial depleted mice**. NIS elements were used to compare a baseline z-stack to z-stacks from time points following complete removal of ablated SR101-labeled astrocytes. Specifically, ablated astrocytes and the optical section(s) their soma occupied were denoted in the baseline image, allowing the surrounding vascular landmarks and astrocytes that were not ablated to serve as fiduciary landmarks when evaluating that same field post-ablation. Any astrocyte at post-ablation timepoints that appeared to extend processes and occupy a vascular vacancy in the baseline image, again using the (1) surrounding vascular profile, (2) astrocytes that were not ablated, and (3) z location of the astrocyte in question relative to the ablated astrocyte. If at baseline, the replacement astrocyte in question did not appear to have processes interacting with the vasculature, even upon dramatically increasing the look up table, this was considered a successful endfoot plasticity response. The frequency of endfoot plasticity events in the absence of microglial cells was then reported.

**Quantification of BBB leakage**. Prior to astrocyte ablation (time point 1), a 30 μL bolus of 100 mg/ml 3 kDa TRITC (D3307 Invitrogen) was retro-orbitally injected, with a timer started at the moment of injection. The mouse was moved immediately to the stereotax under the microscope to capture a z stack, recording the time at the start and end of image capture to enable proper comparison at all subsequent time points. This process was repeated at the moment of fluorescence fading in the process of interest at the vascular interface (time point 2) and the same imaging parameters used as that of the baseline image. FIJI (ImageJ) was used to create sum intensity projections, utilizing the same number of optical sections for all time points, and selecting the optical sections in which the ablated astrocyte was covering the vasculature. Background subtraction was performed using the rolling ball radius feature, and an average fluorescent value was measured at the location of endfoot coverage at both time points. For the induction of vessel injury as a positive control, 870 nm line scans at a laser power of 50–85 mW were applied across the vessel wall for 90–120 s. As in prior BBB measurements, a z stack was captured at the same time after retro-orbital injection of 3 kDa TRITC, and the average intensity of TRITC just beside the damaged vessel was compared at both time points, pre- to post-vessel injury.

**In vivo replacement astrocyte induced-precapillary arteriole constriction-**. To determine if replacement astrocytes can vasoregulate precapillary arterioles, Aldh1l1-cre x GCaMP5G mice received a cranial window following the methodology described above. The first branching capillary segment from Alexa 633 hydrazide-positive penetrating arterioles was selected for imaging if the soma of astrocytes making contact with the precapillary arterioles were on a focal plane similar to the vessel. Using a four-channel Olympus multiphoton laser scanning fluorescence microscope equipped with a XLPLN25X/1.05 NA water-immersion objective (Olympus), single-plane images were obtained every 3 s. Astrocytes were targeted for laser irradiation by selecting a focal plane with both the soma and associated precapillary arteriole visible. Astrocyte stimulation was achieved using a 4 μm$^2$ circular region of interest centered within the astrocyte soma for 800 milliseconds at 7–10× imaging power levels. A one-minute measurement was recorded immediately before and after astrocyte activation. Vessel diameter was measured as the cross-section of the vessel using FIJI (ImageJ) software. Motion correction in videos was performed using the Intravital Microscopy Toolbox ImageJ macro developed by Soulet et al.[69].

**Rose Bengal photothrombosis**. To induce Rose Bengal intravascular clot formation focally and transiently, Aldh1l1-eGFP mice were retro-orbitally injected with Alexa 633 hydrazide. Penetrating arterioles were identified prior to the beginning of the experiment, after which Rose Bengal dye was retro-orbitally injected. All mice <20 g received a 25 μL injection, whereas mice >20 g received a 50 μL injection. Mice were then immediately transferred to the multiphoton, and a square ROI was placed over a penetrating arteriole through the cranial window. The dimensions of the ROI were based upon the dimensions of the penetrating vessel, and imaging was conducted 90 μm from the surface of the brain for two

minutes at a wavelength of 870 nm. Laser power was set between 50 and 90 mW, with power determined on the average intensity of Rose Bengal at the beginning of the experiment. If only part of the vessel was occluded at the end of 2 min, the vessel would be imaged in laser scanning mode until dye nucleation was complete. In all instances, a time period of two minutes was not exceeded. Upon successful intravascular clot formation, a z-stack was captured to visualize the clot, and the animal was placed back in its home cage. The animal was subsequently imaged one hour later to confirm that the clot had cleared. All instances of reported endfoot replacement come from depths below the plane of dye nucleation.

**Drug treatment**. AG490 at 10 mg/kg in 40%DMSO/PBS was subcutaneously injected daily for initial studies comparing percent increase in eGFP volume at day post ablation 5 relative to baseline (Fig. 5). This dosage was increased to 3×/day for studies aiming to prolong the time a vessel region remained vacant post-ablation (Supplementary Fig. 5).

**Pharmacological ablation of microglia using PLX3397**. For microglial depletion studies, mice were fed for eight consecutive days with chow containing a final dose of 660 mg/kg PLX3397, a CSF1R inhibitor widely used to eliminate microglia from the brain[70].

**Tamoxifen injection in Aldh1l1-creERT2 x pSTAT3 fl/fl mice**. Tamoxifen was purchased from Sigma, catalog # T5648. A 20 mg/mL solution was prepared by dissolving tamoxifen first in 10% volume of 100% EtOH, then 90% volume pre warmed (37 °C) corn oil. This solution was then shaken at 270 rpm at 37 °C until fully dissolved. We then subcutaneously injected this solution at 100 mg/kg body weight once/day for five consecutive days. Cranial windows were implanted two weeks following the last injection and imaging commenced one week later.

**Statistics**. GraphPad Prism software was used to perform all statistical analyses. Details for every statistical test are reported in the figure legends. Every parametric test used was validated by first performing tests of normality on the dataset once outliers were removed. Parametric tests were further selected based on datasets having equivalent or different standard deviations, where a difference of <1.5 was counted as being equal. All error bars in bar graphs represent the standard error of the mean.

**Reporting summary**. Further information on research design is available in the Nature Research Reporting Summary linked to this article.

## Data availability
Data can be made available upon request to Harald Sontheimer. Source data are provided with this paper.

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

## Acknowledgements

This work was supported by NIH grants 1R01AG065836 (awarded to HS), 1R01CA227149 (awarded to HS), and 1R01-NS082851 (awarded to H.S). W.A.M. III was supported by the National Institutes of Health Basic Cardiovascular Research Training Grant (5T32HL007284).

## Author contributions

W.A.M. performed all surgeries, designed and performed image acquisition studies, some data analysis, and wrote the manuscript. A.M.W. and S.J. performed data analysis and gave editorial advice. J.M., D.S., and M.B. performed data analysis. I.F.K. helped with experimental design, made representative videos, and gave editorial advice. U.B.E. and M.V.S. provided mice and gave editorial advice. H.S. conceived idea, experimental design, analysis and interpretation, wrote the manuscript, and project supervision.

## Competing interests

The authors declare no competing interests.
