## [Peer Review File · Nature Communications]

Astrocyte plasticity in mice ensures continued endfoot coverage of cerebral blood vessels following injury and declines with ageREVIEWER COMMENTS

Reviewer #1 (Remarks to the Author):

This manuscript is about the replacement of astrocytic endfeet at blood vessels after astrocyte ablation and how that affects the integrity of the blood brain barrier. The authors performed in vivo multiphoton imaging experiments to demonstrate that astrocyte ablation leads to the formation of endfeet on vacated blood vessel by the remaining astrocytes. They also probe vasoconstriction. In addition, they report that the phenomenon is present in animals 4 and 12 month of age but altered in older animals. They further study the role of pSTAT3 pathway and perform reperfusion experiments. Overall, the authors have made an interesting, novel and important observation. However, there are several major and minor concerns. Especially open questions about analysis make an assessment difficult.

Major:

The analyses of experimental results lack depth and are often too qualitative. The authors should provide the percentage of coverage of blood vessels by endfeet before and after ablation and its time course wherever possible. The number of replacement cells (Fig. 3) is not a measure of how much blood vessel surface has been covered by the remaining cells, because process/endfeet growth could be age dependent. A quantitative analysis of the reperfusion experiment (Fig. 6) is necessary for its interpretation.

The experiment presented in Fig. 4 is very interesting. However, in addition to the previous point, it was not entirely clear how this experiment was done. The methods state that animals were imaged twice a day and upon the detection of cell body swelling every five minutes. The spread of data in Fig. 4F (-150 min to +250 min) suggests that they were imaged for many hours every five minutes. Please state the duration of the experiments, the quantitative criteria for cell body swelling and 'fading' of the processes and analyse the delay between both.

There is no quantification of the effect of AG490 on the endfeet replacement process (as suggested above or as presented in Fig. 3 and 4). This needs to be provided. Instead, the authors analyse EGFP volume but there appears to be a technical issue with that (Fig. 5). Astrocytes have been reported to occupy ~ 10% of the tissue volume, for instance in the hippocampus. The DMSO control shows increases to up to ~ 800% of baseline. If both were correct for the studied brain region, that would mean that up to 80% of the tissue volume are occupied by astrocytes after the experiment. This seems unlikely. Also, the exact brain region for experiments and depth below the brain surface need to be stated in the manuscript.

The vasoconstriction assay is not convincing as presented, because a negative control is missing (e.g. same laser exposure but no vasoconstriction).

Minor:

One finding of the study is that extravasation is not affected by astrocyte ablation. Please state in the manuscript how many astrocytes were ablated per experiment (on average). A simple explanation could be that too few astrocytes were ablated to detect an effect.

The colour lookup tables for figures have been adjusted so that primarily cell bodies, large

branches and endfeet of astrocytes and not their fine perisynaptic processes are shown. This should be stated clearly in the manuscript (e.g. figure legend).

How were the authors able to image astrocytes beyond the removal of their 'corpses'? (Methods)

Page 3 and elsewhere: it is not clear what the authors mean by 'innervation/innervate'.

Reviewer #2 (Remarks to the Author):

The authors describe a novel astrocyte response: when an astrocyte contacting a blood vessel in the brain is damaged and its endfeet retract from the vessel, the endfeet of adjacent astrocytes fill in the vacant space on the surface of the vessel. This occurs at arterioles, capillaries and venules. This replacement response occurs rapidly in young animals but is delayed by a few hours in older animals. The loss of endfoot coverage does not, however, result in the breakdown of the blood brain barrier (despite what is implied in the Abstract, see below). The endfoot replacement response is dependent on the STAT3 signaling pathway.

This novel finding is of interest and may have clinical implications, particularly for stroke and microvascular disease.

The Abstract states, "astrocyte plasticity ensures continual vascular coverage so as to retain the BBB". This is a misleading statement, as the authors' results demonstrate that loss of astrocyte coverage, at least for several hours, does NOT disrupt the BBB.

Line 47. Astrocyte-mediated neurovascular coupling is not mediated exclusively by astrocyte purinergic receptors. Other receptors are also involved. Please correct.

Line 244. The authors state that stimulation of replacement endfeet results in vessel constriction. What they fail to report however, is that the magnitude of the constriction appears to be significantly smaller for replacement endfeet (Fig. 2 m,n). In fact, as shown in the figure, laser stimulation evokes similar Ca²⁺ increases in the original and replacement astrocytes, but smaller constrictions for the replacement endfeet, contradicting what they state. Of greater importance, however, is the manner in which the authors stimulate the astrocytes. Laser activation of an astrocyte may damage the cell. This could lead to the non-selective release of ATP, which would constrict vessels. Did the authors insure that laser-stimulated astrocytes were not damaged? Was the constriction repeatable upon repeated stimulation of an astrocyte? Would stimulation of astrocytes not in contact with a vessel also cause constriction? The authors must present evidence that their laser stimulation is not damaging the cells.

It is not clear to this reviewer how a replacement endfoot can be in place on a vessel BEFORE the endfoot of the damaged astrocyte is retracted. How can two endfeet be in the same place at the same time. Please explain exactly what is meant.

Minor comments

The nomenclature used by the authors is inconsistent. They sometimes refer to the first branching vessel off of a penetrating arteriole as a "capillary" and sometimes as a "pre-

capillary arteriole”.

Several acronyms are not defined, including, 2Phatal, TBI, NIS elements.

Line 621. “an astrocyte polarizing a process to the vacant vascular location”. What does polarizing mean?

Note: the figures were reproduced in far too low a resolution in the pdf. I had to download the doc version to look at the figures.

Reviewer #3 (Remarks to the Author):

Manuscript Title: Astrocyte plasticity ensures continued endfoot coverage of cerebral blood vessels and integrity of the blood brain barrier, with plasticity declining with normal aging

Authors: Mills et al.

Using in vivo multiphoton imaging and young and aged Aldh111-eGFP mice and 2Phatal ablation method to induce apoptosis in astrocytes, the authors investigated whether loss of astrocyte endfoot coverage on brain vasculature, including arterioles, venules, and capillaries can be compensated by the neighboring astrocytes. They also used the Rose Bengal photothrombotic transient stroke as an alternative model to induce astrocyte cell death. They found that the time required for endfoot replacement was significantly longer in aged mice than in young mice. They also suggested that the JAK2/STAT3 pathway in astrocytes is essential for focal endfoot replacement response.

While the authors present an interesting and technically challenging study, the extension of the astrocyte endfoot and re-covering of blood vessels after laser ablation has been recently shown by in vivo imaging (PMID: 30718555). Therefore, it would be helpful to emphasize conceptual advances of the present study compared to this previous study. The authors should also consider addressing the following significant concerns.

Comments

1. The JAK/STAT pathway is present in other cell types. Do the authors know whether the activation of the JAK2/STAT3 signaling pathway proposed here to regulate astrocyte endfoot plasticity is self-autonomous or non-autonomous? In other words, is the activation occurs and is restricted to astrocytes or takes place in the neighboring cell types such as endothelium or pericytes and/or other capillary associated cells, which in turn regulate the endfoot replacement. Answering this question will significantly strengthen the novelty.

2. Is aquaporin-4 (AQP4) essential for astrocyte plasticity? Previous studies in AQP4 mice have suggested its role in astrocyte migration (PMID: 16564496)

3. Is pericyte coverage affected with aging and experimental conditions in the studied mouse models? This is important to examine, since pericyte coverage deficits in capillaries could potentially have an effect on retrograde signaling to pre-capillary arterioles that the authors are looking at. Additionally, pericytes have been also shown to control the BBB integrity (PMID: 31235908; PMID: 21040844)

4. Have the authors looked at the microglia activation after focal astrocyte ablation? Microglia can also be implicated in regulation of the BBB permeability especially after injury (PMID: 26755608)

5. In the examples presented, it seems that to achieve territory replacement by neighboring astrocytes that multiple astrocytes in a region must be ablated, instead of a one-to-one type replacement. This is quite different from single pericyte ablation (PMID: 33603231). The authors should better clarify if this is the case, and discuss the implications of loss of multiple astrocytes to prompt an astrocyte to take over the ablated cells territory. Is there a signaling threshold that must be achieved to prompt an astrocyte to take over a territory?

6. Many of the findings are observational. It would strengthen the authors case to provide quantitative comparisons of the original vs replacement calcium fluorescence change and vessel diameter change (figure 2).

7. The authors should provide all methodological details, including the source and genotype of animals used in this study. Please include a statement that the IACUC approved all procedures.

Minor comments:

8. The authors comment at approx. line 320 regarding the difference between a transient ischemic attack and stroke is not clear. Many rodent models of stroke induce ischemic conditions for less than one hour.

9. The authors note that pericytes play a role to regulate capillary vasodilation. The authors should cite relevant literature to support these statements, including Nature. 2014 Apr 3;508(7494):55-60 and Nat Neurosci. 2017 Mar;20(3):406-416.

10. The figures in general could be better presented. Of note, (1) in most cases the difference between the dorsal and dorsolateral views is not obvious. The presentation may be clearer if the authors just chose the view that best illustrates the features of interest. (2) It would also be helpful to indicate which astrocyte will be taking over the territories of the ablated astrocytes in the "before" images.

11. In the methods, a heart rate of 100 beats per minute seems more than "lightly anesthetized." Consider revising.

12. In the methods, it is not clear when the Hoeschst dye is added to the brain. Does this occur during implantation of the cranial window? Or do these experiments occur in mice with acute window preps?

13. In the methods, information on z-stack parameters is missing.

14. While cadaverine did not work well in this experiment, it would be informative to indicate which specific cadaverine was used. The molecular weight, and potentially properties, of cadaverine vary depending on the fluorescent molecule to which it was linked.

15. Figure 5: In panels b, c, it is not clear that the astrocytes were ablated. Please label the conditions for figure 5g and h. Also, was there any statistical difference between the post-

ablation conditions for vehicle and AG490 conditions?

16. Figure 6f-i: The astrogliosis present in this condition makes it very difficult to make out the features the authors are trying to illustrate. Please revise.

Point-by-point list of changes

Reviewer #1 (Remarks to the Author):

This manuscript is about the replacement of astrocytic endfeet at blood vessels after astrocyte ablation and how that affects the integrity of the blood brain barrier. The authors performed in vivo multiphoton imaging experiments to demonstrate that astrocyte ablation leads to the formation of endfeet on vacated blood vessel by the remaining astrocytes. They also probe vasoconstriction. In addition, they report that the phenomenon is present in animals 4 and 12 month of age but altered in older animals. They further study the role of pSTAT3 pathway and perform reperfusion experiments. Overall, the authors have made an interesting, novel and important observation. However, there are several major and minor concerns. Especially open questions about analysis make an assessment difficult.

Major:

1. The analyses of experimental results lack depth and are often too qualitative. The authors should provide the percentage of coverage of blood vessels by endfeet before and after ablation and its time course wherever possible. The number of replacement cells (Fig. 3) is not a measure of how much blood vessel surface has been covered by the remaining cells, because process/endfeet growth could be age dependent. A quantitative analysis of the reperfusion experiment (Fig. 6) is necessary for its interpretation.

Response: We agree with this critique and thank the reviewer for highlighting this shortcoming. We have provided a quantitative comparison from Aldh1l1-eGFP mice of the eGFP+ area around penetrating arterioles before and after ablation. We highlight in this comparison that astrocyte morphological association with the vasculature can be broken down into two categories: those in which the soma and associated processes are on the vessel, and those in which the soma lies in parenchymal space with a clear process extended towards the vasculature. We refer to the former as vessel-associated astrocytes (VAA) and the latter as parenchymal-associated astrocytes (PAA). See Supplementary Figure 4a-c for images conveying this categorization. eGFP+ area was significantly reduced post-ablation in locations of VAA ablation. This is predictable as astrocytes do not move their somas in response to injury (see reference 41), and our observations show this is consistent for ablations too. Hence, replacing an astrocyte soma and processes with only processes from surrounding astrocytes will result in significantly reduced eGFP+ area, and for this reason, while replacement astrocyte number is not sufficient when reported alone, it does provide useful information. For PAA ablations, there was a small yet significant difference in post-ablation coverage relative to baseline, where post-ablation coverage was roughly 95% of baseline values. Nearly maintaining baseline levels of endfoot vessel coverage makes sense as you are replacing a process with processes from a surrounding astrocyte. This data is provided in Supplementary Figure 4, and we focused our remaining analyses on PAA astrocytes only. The one exception is the pre-existing AG490 drug studies we provided in the initial draft of this manuscript (see comment 3). A graphical presentation of our analysis paradigm is also provided in Supplementary Figure 8. As you will read in comment two, providing the exact time course of replacement requires very long imaging sessions and hence is not feasible for every experiment. However, we have managed to keep the day of post-ablation measurements the same for all comparisons made. Regarding the reperfusion experiment, we simply wanted to show that this process occurs in this paradigm. We are currently conducting additional studies for a future manuscript to further elucidate the role of endfoot plasticity in vessel recovery following transient ischemic attack, part of which will be more thoroughly characterizing the reperfusion experiment.

2. The experiment presented in Fig. 4 is very interesting. However, in addition to the previous point, it was not entirely clear how this experiment was done. The methods state that animals were imaged twice a day and upon the detection of cell body swelling every five minutes. The spread of data in Fig. 4F (-150 min to +250 min) suggests that they were imaged for many hours every five minutes. Please state the duration of the experiments, the quantitative criteria for cell body swelling and 'fading' of the processes and analyze the delay between both.

Response: Our apologies for the confusion regarding this experiment. We have added the duration of each imaging session (8-12 hours), the quantitative criteria for cell body swelling and fading of processes to the methods section. There is not a significant difference in the delay between cell body swelling and processes fading (see figure below).

Delay between cell body swelling and processes fading

Two-tailed unpaired t-test, n=5 cells/3 mice for both groups, $p < 0.6205$

3. There is no quantification of the effect of AG490 on the endfeet replacement process (as suggested above or as presented in Fig. 3 and 4). This needs to be provided. Instead, the authors analyze EGFP volume but there appears to be a technical issue with that (Fig. 5). Astrocytes have been reported to occupy ~ 10% of the tissue volume, for instance in the hippocampus. The DMSO control shows increases to up to ~ 800% of baseline. If both were correct for the studied brain region, that would mean that up to 80% of the tissue volume are occupied by astrocytes after the experiment. This seems unlikely. Also, the exact brain region for experiments and depth below the brain surface need to be stated in the manuscript.

Response: This is an excellent point brought up by the reviewer. We are confident in the thresholding of our images as we set reference astrocytes in each field of view at baseline when setting up laser power and PMT values in high-low mode. These same astrocytes were used at post-ablation timepoints to ensure that the histogram was set up as nearly identical as possible. Given the small number of astrocytes ablated, it is unlikely that replacement astrocytes are actually occupying 80% of tissue volume though the raw values might suggest that. However, to more thoroughly and specifically analyze how loss of STAT3 phosphorylation may impact endfoot plasticity, we have included a new experiment from *Aldh1l1creERT2 x pSTAT3 fl/fl* mice where pSTAT3 phosphorylation is exclusively prevented in astrocytes following tamoxifen-induced Cre activation. We not only assess individual replacement astrocyte volume, but we assess eGFP+ endfoot replacement area at the vessel following ablation (See Figure 6g). We went with this analysis based on your comments here and in comment 1, and we appreciate this critique. Finally, we have added the exact brain region and depth below the surface for experiments to the methods section in the manuscript. Specifically, we imaged between 100 and 200 μ m below the cortical surface in somatosensory cortex.

4. The vasoconstriction assay is not convincing as presented, because a negative control is missing (e.g. same laser exposure but no vasoconstriction).

Response: We now provide control data including a representative negative control for the figure showing equal stimulation yet no constriction of the vessel. Importantly, the control astrocytes are by the vessel but have no apparent processes occupying vascular territories, thereby showing the necessity of vascular interaction by processes to induce constriction upon stimulation.

Minor:

5. One finding of the study is that extravasation is not affected by astrocyte ablation. Please state in the manuscript how many astrocytes were ablated per experiment (on average). A simple explanation could be that too few astrocytes were ablated to detect an effect.

Response: Our apologies for excluding this information, and this is an interesting point that we address in the discussion.

6. The colour lookup tables for figures have been adjusted so that primarily cell bodies, large branches and endfeet of astrocytes and not their fine perisynaptic processes are shown. This should be stated clearly in the manuscript (e.g. figure legend).

Response: This has been added at the end of appropriate figure legends.

7. How were the authors able to image astrocytes beyond the removal of their 'corpses'? (Methods)

Response: We have clarified this point in the methods section

8. Page 3 and elsewhere: it is not clear what the authors mean by 'innervation/innervate'.

Response: We have added a sentence where we first introduce the concept of endfoot plasticity as that in which astrocytes extend processes to a vacant vascular region, thereby innervating it. We have done our best to be consistent with this language.

Reviewer #2 (Remarks to the Author):

The authors describe a novel astrocyte response: when an astrocyte contacting a blood vessel in the brain is damaged and its endfeet retract from the vessel, the endfeet of adjacent astrocytes fill in the vacant space on the surface of the vessel. This occurs at arterioles, capillaries and venules. This replacement response occurs rapidly in young animals but is delayed by a few hours in older animals. The loss of endfoot coverage does not, however, result in the breakdown of the blood brain barrier (despite what is implied in the Abstract, see below). The endfoot replacement response is dependent on the STAT3 signaling pathway.

This novel finding is of interest and may have clinical implications, particularly for stroke and microvascular disease.

1. The Abstract states, "astrocyte plasticity ensures continual vascular coverage so as to retain the BBB". This is a misleading statement, as the authors' results demonstrate that loss of astrocyte coverage, at least for several hours, does NOT disrupt the BBB.

Response:

We removed the reference to endfoot coverage ensuring BBB integrity in the revised Abstract and Manuscript Title

2. Line 47. Astrocyte-mediated neurovascular coupling is not mediated exclusively by astrocyte purinergic receptors. Other receptors are also involved. Please correct.

Response: We have corrected this in our introduction section

3. Line 244. The authors state that stimulation of replacement endfeet results in vessel constriction. What they fail to report however, is that the magnitude of the constriction appears to be significantly smaller for replacement endfeet (Fig. 2 m,n). In fact, as shown in the figure, laser stimulation evokes similar Ca²⁺ increases in the original and replacement astrocytes, but smaller constrictions for the replacement endfeet, contradicting what they state. Of greater importance, however, is the manner in which the authors stimulate the astrocytes. Laser activation of an astrocyte may damage the cell. This could lead to the non-selective release of ATP, which would constrict vessels. Did the authors insure that laser-stimulated astrocytes were not damaged? Was the constriction repeatable upon repeated stimulation of an astrocyte? Would stimulation of astrocytes not in contact with a vessel also cause constriction? The authors must present evidence that their laser stimulation is not damaging the cells.

Response: This is an excellent point brought up by the reviewer and we thank them for doing so. We have added a negative control showing lack of constriction upon laser-stimulation of astrocytes just beside a vessel but lacking visible processes occupying vascular territory. Unfortunately, we were not able to repeatedly stimulate astrocytes to

induce vessel constrictions. This is also the case for those astrocytes in our negative control. We interpret these results to suggest that process interaction is necessary to induce a vessel response, albeit in this case, it could be by ATP release from a dying cell. The benefit of using this stimulation method is that one can definitively attribute immediate vessel responses following stimulation to the cell that was directly stimulated, which is why we chose this method.

4. It is not clear to this reviewer how a replacement endfoot can be in place on a vessel BEFORE the endfoot of the damaged astrocyte is retracted. How can two endfeet be in the same place at the same time. Please explain exactly what is meant.

Response: This wording was unfortunate and has been altered. We intended to express that the replacement endfoot is in place and ready to take over practically immediately upon ablation of the original.

Minor comments

5. The nomenclature used by the authors is inconsistent. They sometimes refer to the first branching vessel off of a penetrating arteriole as a "capillary" and sometimes as a "pre-capillary arteriole".

Response: This was clarified. We exclusively used pre-capillary arterioles and have referred to all as such.

6. Several acronyms are not defined, including, 2Phatal, TBI, NIS elements.

Response: We have now defined all of these acronyms in the manuscript. 2Phatal: Two-photon TBI: Traumatic Brain Injury. chemical apoptotic ablation. NIS elements: Nikon Imaging Software Elements

7. Line 621. "an astrocyte polarizing a process to the vacant vascular location". What does polarizing mean?

Response: We meant 'extend' and we have removed this word from the manuscript. Thank you for bringing this to our attention.

8. Note: the figures were reproduced in far too low a resolution in the pdf. I had to download the doc version to look at the figures.

Response: We have enlarged each figure and have provided them in much higher resolution. Our apologies for this inconvenience.

Reviewer #3 (Remarks to the Author):

Manuscript Title: Astrocyte plasticity ensures continued endfoot coverage of cerebral blood vessels and integrity of the blood brain barrier, with plasticity declining with normal aging

Authors: Mills et al.

Using in vivo multiphoton imaging and young and aged Aldh111-eGFP mice and 2Phatal ablation method to induce apoptosis in astrocytes, the authors investigated whether loss of astrocyte endfoot coverage on brain vasculature, including arterioles, venules, and capillaries can be compensated by the neighboring astrocytes. They also used the Rose Bengal photothrombotic transient stroke as an alternative model to induce astrocyte cell death. They found that the time required for endfoot replacement was significantly longer in aged mice than in young mice. They also suggested that the JAK2/STAT3 pathway in astrocytes is essential for focal endfoot replacement response.

While the authors present an interesting and technically challenging study, the extension of the astrocyte endfoot and re-covering of blood vessels after laser ablation has been recently shown by in vivo imaging (PMID: 30718555). Therefore, it would be helpful to emphasize conceptual advances of the present study compared to this previous study. The authors should also consider addressing the following significant concerns.

Comments

1. The JAK/STAT pathway is present in other cell types. Do the authors know whether the activation of the JAK2/STAT3 signaling pathway proposed here to regulate astrocyte endfoot plasticity is self-autonomous or non-autonomous? In other words, is the activation occurs and is restricted to astrocytes or takes place in the neighboring cell types such as endothelium or pericytes and/or other capillary associated cells, which in turn regulate the endfoot replacement. Answering this question will significantly strengthen the novelty.

Response: To get at this question we entered a collaboration with Michael Sofroniew using the Aldh111-creERT2 x pSTAT3 fl/fl mice that his laboratory generated several years ago. Here, we injected tamoxifen for five consecutive days, implanted cranial window two weeks later, and began imaging one week after that. Measuring individual astrocyte volume following astrocyte ablation revealed that replacement astrocytes in the control group (Cre- mice) significantly increased their volume from baseline levels relative to Cre+ mice. We further show that eGFP+ area around the vessel remains constant in control mice, whereas there is roughly a 50% reduction in experimental mice. This data suggests that STAT3 signaling in astrocytes is indeed the culprit.

2. Is aquaporin-4 (AQP4) essential for astrocyte plasticity? Previous studies in AQP4 mice have suggested its role in astrocyte migration (PMID: 16564496)

Response: This is an excellent point provided by this reviewer. Unfortunately, the genetic tools to conduct this experiment were not accessible to us, but we have included this point in our discussion.

3. Is pericyte coverage affected with aging and experimental conditions in the studied mouse models? This is important to examine, since pericyte coverage deficits in capillaries could potentially have an effect on retrograde signaling to pre-capillary arterioles that the authors are looking at. Additionally, pericytes have been also shown to control the BBB integrity (PMID: 31235908; PMID: 21040844)

Response: This is an excellent point. We do not observe any pericyte deficits following focal ablation of astrocytes as evidenced by NG2 volume measurements and capillary diameter measurements (Supplementary Figure 2).

It is well known that pericyte loss occurs in normal biological aging. Numerous groups have shown this, including in human post-mortem samples (see reference below). We immunostained for CD13 in 3-month and 12-month C57 mice and quantified pericyte volume on capillaries as a percentage of total laminin-111. We show a significant reduction in CD13 (see figure below). Since we did not measure BBB integrity at capillaries in aging following astrocyte ablation, however, we have decided to exclude this data from the current version of the manuscript. $p < 0.0001$, unpaired t-test, $n = 23$ capillaries/5 mice for each age group.

Ding R., Hase Y., Burke M. *et al.* Loss with ageing but preservation of frontal cortical capillary pericytes in post-stroke dementia, vascular dementia and Alzheimer's disease. *acta neuropathol commun* 9, 130 (2021).
<https://doi.org/10.1186/s40478-021-01230-6>

4. Have the authors looked at the microglia activation after focal astrocyte ablation? Microglia can also be implicated in regulation of the BBB permeability especially after injury (PMID: 26755608)

Response: We agree with the reviewer that this was necessary to investigate and thus collaborated with Ukpong Eyo to address this question through additional experiments. We now include data from CX3CR1-eGFP mice where we ablated SR101-labeled astrocytes. We see that microglia are recruited to the site of ablation and phagocytose 100% of ablated astrocytes. When these experiments were performed in P2RY12 KO mice, there was no change in microglia engulfment by astrocytes, suggesting that it occurs independent of purinergic signaling. Finally, we pharmacologically ablated microglia using PLX3397 and show that astrocyte plasticity remains intact following astrocyte ablation, suggesting that recruitment of replacement astrocytes is not a microglia-directed phenomenon.

5. In the examples presented, it seems that to achieve territory replacement by neighboring astrocytes that multiple astrocytes in a region must be ablated, instead of a one-to-one type replacement. This is quite different from single pericyte ablation (PMID: 33603231). The authors should better clarify if this is the case, and discuss the implications of loss of multiple astrocytes to prompt an astrocyte to take over the ablated cells territory. Is there a signaling threshold that must be achieved to prompt an astrocyte to take over a territory?

Response: This is an excellent point brought up by the reviewer. We have provided a quantitative comparison from Aldh111-eGFP mice of the eGFP+ area around penetrating arterioles before and after single astrocyte ablation. We highlight in this comparison that astrocyte morphological association with the vasculature can be broken down into two categories: those in which the soma and associated processes are on the vessel, and those in which the soma lies in parenchymal space with a clear process extended towards the vasculature. We refer to the former as vessel-associated astrocytes (VAA) and the latter as parenchymal-associated astrocytes (PAA) (see Supplementary Figure 4a-c for images showing an example of each). Results show a one-to-one type replacement, albeit eGFP+ area is significantly reduced post-ablation for ablation of VAA and PAA astrocytes. The average post-ablation eGFP+ area at arterioles for PAA ablations was roughly 95% of baseline and VAA ablations 45% of baseline. This data is provided in Supplementary Figure 4. Additionally, we highlight in our discussion the concept of ablation size (or signaling threshold) that is needed to get a replacement response versus no response.

6. Many of the findings are observational. It would strengthen the authors case to provide quantitative comparisons of the original vs replacement calcium fluorescence change and vessel diameter change (figure 2).

Response: This is an excellent critique, and in addition to what we stated in the above response to comment number 5, we have added further quantitative analyses of the original vs replacement calcium fluorescence change and vessel diameter change.

7. The authors should provide all methodological details, including the source and genotype of animals used in this

study. Please include a statement that the IACUC approved all procedures.

Response: Our apologies to this reviewer for a lack of clarity on IACUC approval. We have included a statement confirming this. We have further provided clarity on all methodological details, including the source and genotype of animals used in the study.

Minor comments:

8. The authors comment at approx. line 320 regarding the difference between a transient ischemic attack and stroke is not clear. Many rodent models of stroke induce ischemic conditions for less than one hour.

Response: This is an excellent point, and we apologize for the lack of clarity. We emphasize that we are using the clinical definition based on time to reperfusion.

9. The authors note that pericytes play a role to regulate capillary vasodilation. The authors should cite relevant literature to support these statements, including Nature. 2014 Apr 3;508(7494):55-60 and Nat Neurosci. 2017 Mar;20(3):406-416.

Response: These citations have been added to our manuscript. See references 37 and 38

10. The figures in general could be better presented. Of note, (1) in most cases the difference between the dorsal and dorsolateral views is not obvious. The presentation may be clearer if the authors just chose the view that best illustrates the features of interest. (2) It would also be helpful to indicate which astrocyte will be taking over the territories of the ablated astrocytes in the “before” images.

Response: We have chosen to use the dorsolateral view for presentation in our new additional experiments, and we indicate which astrocytes will be taking over the territory of the ablated astrocyte with arrow heads in DPA0 images. This is outlined in each figure legend too. We thank the reviewer for the suggestion.

11. In the methods, a heart rate of 100 beats per minute seems more than “lightly anesthetized.” Consider revising.

Response: We have revised our methods section accordingly

12. In the methods, it is not clear when the Hoeschst dye is added to the brain. Does this occur during implantation of the cranial window? Or do these experiments occur in mice with acute window preps?

Response: We have revised our methods section to provide further clarity. Hoescht dye was added in a 10-minute incubation step after a durotomy was performed but prior to placing the 3x3mm window. This is consistent with what Hill et al did in the original 2Phatal paper (reference 31 in our manuscript)

13. In the methods, information on z-stack parameters is missing.

Response: This information has been added to the methods section

14. While cadaverine did not work well in this experiment, it would be informative to indicate which specific cadaverine was used. The molecular weight, and potentially properties of, cadaverine vary depending on the fluorescent molecule to which it was linked.

Response: We have included the catalogue number for the specific Cadaverine we used.

15. Figure 5: In panels b, c, it is not clear that the astrocytes were ablated. Please label the conditions for figure 5g and h. Also, was there any statistical difference between the post-ablation conditions for vehicle and AG490 conditions?

Response: We have made the white asterisks in panels b and c larger and more robust in appearance. The conditions for figure 5g and h have been labeled. 5g quantifies BBB results from AG490 mice and 5h from DMSO injected control mice.

As shown in the figure below, we do not see a significant difference between the post-ablation conditions for vehicle and AG490 injected mice. n=16 arterioles/4 mice, Two-tailed unpaired t-test, p < 0.2331

AG490 versus DMSO Post-Ablation BBB Results

16. Figure 6f-i: The astrogliosis present in this condition makes it very difficult to make out the features the authors are trying to illustrate. Please revise.

Response: We have completely revised this figure in an effort to make interpretation of the results more accessible

REVIEWER COMMENTS

Reviewer #1 (Remarks to the Author):

The authors have fully addressed my previous comments. They describe a novel and important phenomenon that could be highly relevant for translational research.

Reviewer #2 (Remarks to the Author):

The authors have satisfactorily revised the manuscript. They have addressed all of my concerns.

Reviewer #3 (Remarks to the Author):

The authors have adequately addressed the comments. No further comments.